# Blackbird language matrices (BLM), a new task for rule-like generalization in neural networks: Can Large Language Models pass the test?

**Paola Merlo**

University of Geneva

Paola.Merlo@unige.ch

## Abstract

How do we evaluate LLMs and determine the aspects and limits of their intelligent behaviour?

When exposed to visual tests of analytic intelligence, human problem-solvers identify rules applied to relevant objects and attributes. Based on the induced rules, they can generalise and are able to provide a solution to the test.

An analogous language task has recently been proposed (called BLM) for LLM. In this paper, we use this task to investigate what linguistic reasoning LLM develop, by asking them to solve some simple variants of the BLM task.

We find that current state-of-the-art generative models can handle the task: they easily understand the instructions and can provide step-by-step explanations. The explanations show that LLMs can solve two of the main hurdles: correspondence finding (object and attribute identification) and item novelty. However, overall they struggle to find the correct underlying global rules, even when they find the right answer.

We argue that these findings support the usefulness of the task as a method to test the limits and specific properties of generalisation ability in Large Language Models, providing an intrinsic evaluation method inspired by tests of human intelligence.

## 1 Prologue

How do we evaluate LLMs, determine the aspects and limits of their intelligent behaviour and compare them to human intelligence?

A new trend of large-scale multi-task LLM and neural network evaluation has emerged that aims to evaluate LLM in realistic settings. In NLP we see Big-bench initiatives, to better chart LLMs capabilities (Srivastava et al., 2023). In neuroscience, a similar trend, dubbed integrative bench-marking, looks at large language models as mechanistic hypotheses on how the brain encodes intelligence, models that can be tested on behavioural and neural data (Schrimpf et al., 2020). Finally, another point of view has been put forth recently that, to further understand the components of human intelligence, one needs to study how this intelligence evolved, decomposing it into large-scale studies of primitive skills that humans share with other animals (Zador et al., 2023).

Large scale, highly integrative, comparative approaches are necessary: they give a bird's eye view essential to map unknown territories, define boundaries, guide long-term trends. But we also think that curated data and in-depth qualitative analyses of the intelligent behaviour of these models are needed (Rogers, 2021).

We will concentrate on one such skill, one of the core intelligent human abilities, the ability for analytic rule-based generalisation. We will illustrate what tests exists for this skill and evaluate qualitatively how GPTs rate on that, based on carefully curated data.

## 2 Rule-based generalisation

Recent approaches to generalisation have shifted attention from the traditional narrow definition — an extension from a set of data points to new data points of exactly the same nature (i.i.d. assumption) (Schölkopf, 2019)— to out-of-distribution generalisation, be it across languages, across domains, or new unseen combinatorial tokens and structures (Hupkes et al., 2022).

Humans are good generalizers. A large body of literature of experimental work has demonstrated that the human mind is predisposed to extract regularities and infer the generative rules from data, in a way that is distinct from the patterns of activation of neural networks (Lakretz et al., 2019, 2021; Sablé-Meyer et al., 2021; Dehaene et al., 2022).

One possible approach to develop more robust methods, then, is to pay more attention to the decomposition of complex observations and to the

causal chains in the generative process that gives rise to the data. To discover the underlying problem structure, the notion of disentanglement comes to the rescue. A disentangled representation can be defined as one where single latent units are sensitive to changes in single generative factors, while being relatively invariant to changes in other factors (Bengio et al., 2013).

Let's look at an illustrative example of complex linguistic relations that might require discovering the underlying disentangled factors and causal relations: the CAUSATIVE alternation exhibited by English verbs, shown in (1).

(1)   The teacher   opened   the door.
      AGENT                  THEME
      The door      opened.
      THEME

This alternation applies to change of state verbs, such as *open, break, melt, burn* among many others, verbs that describe a change that affects the state of the undergoing participant (*the door* changes from a state of being closed to a state of being open). They occur in two subcategorisation frames that are related to each other in a regular way: the object of the transitive frame is the subject of the intransitive frame. This way, in terms of semantic roles, the subject of the transitive is an AGENT, but the subject of the intransitive is a THEME.

To learn the structure of such a complex alternation automatically, despite the simplicity of the sentences, a neural network must be able to identify and solve a set of subproblems and compose them into an organised solution: (i) identify the elements manipulated by the alternation (teacher, door); (ii) identify the relevant attributes of these elements (open, closed state of door; volition of teacher); (iii) the object of the transitive sentence is the same event participant as the subject of intransitive, that is the rules of change of the attributes must be identified across the two sentences.

To study what factors and models lead to learning more disentangled linguistic representations, representations that reflect the underlying linguistic rules of grammar, a possible approach is to develop curated data on a large scale, building models to learn these data and investigating the models' behaviour.

To this end, a new linguistic task has been developed, (similar to the IQ test called RPM, Raven's

progressive matrices for vision), called Blackbird Language Matrices (BLMs) (Merlo, 2023). BLMs define prediction tasks to learn complex linguistic patterns and paradigms.

Because this task relies on a mixture of language and abstract rule learning abilities, we conjecture that, used in an evaluation setting, it is appropriate to investigate the limits and properties of linguistic intelligence and abilities in current state-of-the-art LLM, and compare them to humans.

## 3   How do humans solve RPMs?

Raven's progressive matrices are a visual test of analytic intelligence (Raven, 1938). To establish how to develop human-like rule-based inference, it pays to survey how humans solve this kind of intelligence test.

### 3.1   Raven's Progressive Matrices for vision

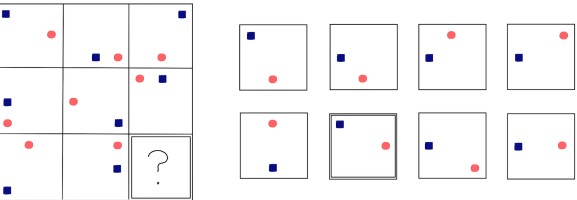

Figure 1: Example of progressive matrice in the visual world. The task is to determine the missing element in a visual pattern. Given the matrix on the left, choose the last element of the matrix from the choice of elements on the right. The matrix is constructed according to two rules (see text for explanation). Identifying these rules leads to the correct answer (marked by double edges).

Raven's progressive matrices (progressive because tasks get harder) are IQ tests consisting of a sequence of images, usually organised in a 3x3 matrix, called the *context*, connected in a logical sequence by underlying generative rules (Raven, 1938). The task is to determine the missing element, usually the last one, in this visual sequence, the *answer*. The answer is chosen among a set of closely or loosely similar alternatives. A simple instance of this task, of our devising, is given in Figure 1: given a matrix on the left, choose the last element of the matrix from a choice of elements. The matrices are built according to generative rules that span the whole sequence of stimuli and the answers are constructed to be similar enough that the solution can be found only if the rules are identified correctly. For example in Figure 1, the matrix is

constructed according to two rules: Rule 1: from left to right, the red dot moves one place clockwise each time. This pattern continues onto the next row; Rule 2: from top to bottom, the blue square moves one place anticlockwise each time. This pattern continues onto the next column. Identifying these rules leads to the correct answer, the only cell that continues the generative rules correctly. Variants of this structure organise the rules row by row, with the last element of the last row as the missing element.

### 3.2 Measuring analytic intelligence in humans

In a very articulated study, Carpenter et al. (1990) analyze the cognitive processes at play in Raven Progressive matrices tests (RPMs), as a widely used, nonverbal test of analytic intelligence. The analysis is based on detailed performance characteristics such as think-aloud verbal protocols, eye fixation patterns, and errors. Of particular interest for us are the three main sources of difficulty that are identified by the study: the correspondence problem, goal and subgoal management, and item novelty.

**Finding corresponding elements** In problems with multiple generative rules, the problem-solver must determine which visual elements or attributes in the entries are governed by the same rule, a process that Carpenter and colleagues call *correspondence finding*. Interestingly, it is conjectured that even for visual problems, people use a linguistic method to find correspondences. Namely, the correspondences reflect the use of a matching-names heuristic: figures with the same name might correspond to each other.

**Goal and subgoal management** In recordings of eye movements and verbal protocols, the researchers found signs of a demonstrably incremental nature of the processing. The subjects solve a problem by decomposing it into successively smaller sub-problems and then proceed to solve each sub-problem. This brought about an incremental, and sometimes recursive, solution to the problem. The induction of the rules was incremental in two respects. First of all, in problems containing more than one rule, the rules were described one at a time. Second, the induction of a rule consisted of many small steps, reflected in the pairwise comparison of elements in adjoining entries. Moreover, the verbal reports show that the subject mentioned one attribute at a time and also

one rule at a time. The general result, then, is that the processing is incremental, that the number of rule tokens affects the error rates, and that there is a wide range of differences among individuals in their performance on this test.

**Item Novelty** Variety and novelty of lexical items was also found to be a source of difficulty.

In conclusion, the study finds that the processing characteristic that is common to all subjects is an incremental, re-iterative strategy for encoding and inducing the regularities in each problem. The processes that distinguish among individuals are primarily the ability to induce abstract relations and the ability to dynamically manage a large set of problem-solving goals in working memory. These two abilities are what distinguish, according to this study, better problem-solvers from average ones.

## 4 Blackbird Language Matrices

| | CONTEXTS TEMPLATE | | | |
|---|---|---|---|---|
| 1 | NP-sg | PP1-sg | | VP-sg |
| 2 | NP-pl | PP1-sg | | VP-pl |
| 3 | NP-sg | PP1-pl | | VP-sg |
| 4 | NP-pl | PP1-pl | | VP-pl |
| 5 | NP-sg | PP1-sg | PP2-sg | VP-sg |
| 6 | NP-pl | PP1-sg | PP2-sg | VP-pl |
| 7 | NP-sg | PP1-pl | PP2-sg | VP-sg |
| 8 | NP-pl | PP1-pl | PP2-sg | VP-pl |
| | ANSWER SET | | | |
| 1 | NP-sg PP1-sg et NP2 VP-sg | | | Coord |
| 2 | **NP-pl PP1-pl NP2-sg VP-pl** | | | correct |
| 3 | NP-sg PP1-sg VP-sg | | | WNA |
| 4 | NP-sg PP1-sg PP2-sg VP-pl | | | AE |
| 5 | NP-pl PP1-sg PP1-sg VP-pl | | | WN1 |
| 6 | NP-pl PP1-pl PP2-pl VP-pl | | | WN2 |

Figure 2: BLM instances for verb-subject agreement, with two attractors. WNA=wrong nr. of attractors; AE=agreement error; WN1=wrong nr. for $1^{st}$ attractor (N1); WN2=wrong nr. for $2^{nd}$ attractor (N2).

We propose to apply a similar test to a large language model. We use here the new task and data format, called BLACKBIRD'S LANGUAGE MATRICES (BLMs), proposed in Merlo et al. (2022) and defined formally in Merlo (2023). (See also An et al. (2023); Samo et al. (2023); Merlo et al. (2023a) for more details on data creation for grammar problems and publicly available data).

Like RPMs, a BLM instance consists of a context set and an answer set. The context is a sequence of sentences that encode a linguistic rule - e.g., subject and verb number should agree in their number independently of how many noun phrases intervene between them. BLMs are presented (to

an artificial language model or to a human partici-pant) as linguistic puzzles requiring the selection of the missing sentence. In order to examine the repre-sentations underlying the response, the answer sets include negative examples following corrupted gen-erating rules. An example template is illustrated in Figure 2.

For example, the BLM problem for subject-verb agreement (Merlo et al., 2022; An et al., 2023) con-sists of a context set of seven sentences that share the subject-verb agreement phenomenon, but dif-fer in other aspects – e.g. number of intervening attractors between the subject and the verb, differ-ent grammatical numbers for these attractors, and different clause structures. The answer set com-prises contrastive sentences that violate some of the generative rules. Actual examples can be found in Section 5.

## 4.1 The linguistic phenomena

The linguistic phenomena we have tested are subject-verb number agreement, the causative al-ternation and the sequence of tenses. In a BLM, the linguistic phenomena have two levels of prop-erties: the external properties (E), those that are easily seen in the strings, and internal properties (I), that are only inferrable by abstraction (see for-mal specification in Merlo (2023) and Merlo et al. (2023b)).

## 4.2 The agreement matrix

The prompt exemplified in Figure 4 shows an ex-ample of a BLM agreement matrix. In subject-verb agreement the primary property is that the subject and the verb must match in grammatical number (singular and plural). A secondary property indi-cates that these two elements do not need to be linearly adjacent, because the rule applies on the syntactic structure of the sentence and not on its linear order (Gulordava et al., 2018; Linzen et al., 2016). In this respect, intervening elements can end up having the effect of confounding the agreement pattern.

The sequence is generated by a rule of progres-sion of number of attractors (one and two), a rule of subject-verb agreement that alternates every sen-tence between singular and plural of the head noun and a rule of number of the attractors that alter-nates between singular and plural with a frequency period of two. Thus, the correct answer for this example is a sentence that has three noun phrases

and a plural subject and plural first attractor and singular second attractor.

## 4.3 The causative matrix

The linguistic specification of the causative alterna-tion has been given in Section 2. The structure of the causative matrix, shown in the prompt exempli-fied in Figure 5, is different from the previous one and it is both harder and easier than the previous one. It is easier in that we use a property of easy progressive matrices, namely the sequence of stim-uli is organised into subsets. And it is harder be-cause the rule to be learnt is more implicit and has no surface morphological correlate, unlike agree-ment.

The first four-tuple of sentences is meant to in-dicate that the subject of the transitive is agentive (first sentence), that the subject of the intransitive is not necessarily agentive (second sentence), and the next two sentences express the initial and final state of the change of state. The second quadru-plet, of which the last element needs to be guessed, repeats the same pattern. The sequence pattern, then, is quite easy to see. But the underlying rules to be learnt have to do with the mapping of gram-matical functions to semantic roles (subject=Agent, object=theme in the transitive, subject=theme in the intransitive), a notoriously difficult task in neu-ral networks.

## 4.4 The sequence-of-tense matrix

In this matrix, we try a different kind of long-distance agreement problem, the so called sequence-of-tenses rule. The sequence of tenses rule governs the coherent use of tenses in different sentences. It is especially used in hypotheticals, which require subjunctive and conditional mood, in the present, past and future tenses. The language is Italian, so that the verb conjugation is clearly marked morphologically, both in mood and tense, also making a difference between singular and plu-ral.

## 4.5 The mixture matrix

Finally, we try a more complex case, in which we have both the sequence of tense rule and the agree-ment rule, which in this case covers both number and gender agreement. The pattern template of the data is shown in Figure 3. An example is shown in the prompt of Figure 7. As can noticed, the agree-ment pattern is similar to the pattern shown in Fig-ure 2, with number agreement alternating at every

sentence (Sg,Pl,Sg,Pl,..), the gender agreement pattern alternating every two sentences (M,M,F,F,...) and an intervening attractor, exhibiting no agreement pattern, which is absent in the first four sentences and present in the last four sentences. Notice that, in this matrix, the attractor is an incidental sentence, so it also interferes in structure and distance with the sequence of tenses, by introducing a third verb and lengthening the distance between the elements in the tense sequence.

| | CONTEXT | | |
|---|---|---|---|
| 1 | SgM-CPas | | SgM-CnPr |
| 2 | PlM-CPas | | PlM-CnPr |
| 3 | SgF-CTrPas | | SgM-CnPas |
| 4 | PlF-CTrPas | | PlM-CnPas |
| 5 | SgM-IFut | attractorPLF | SgM-IFut |
| 6 | PlM-IFutAnt | attractorSgM | PlM-IFut |
| 7 | SgF-IFut | attractorPlM | SgF-IFut |
| 8 | ?? | | |
| | ANSWERS | | |
| A | PlF-IFutAnt | attractorSgF | PlF-IFut |
| B | SgF-IFut | | SgF-IFut |
| C | SgM-CTrPas | | SgM-CnPas |
| D | PlM-IFutAnt | | PlM-IFut |
| E | SgF-CTrPas | attractorSgF | PlM-CTrpas |
| F | PlF-IFutAnt | | PlF-IFut |

Figure 3: Contexts and Answers patterns of the mixture BLM (CPas:past subjunctive; CTrPas:perfect past subjunctive; CnPr:present conditional; CnPas:past conditional; IFut:future indicative; IFutAnt:future in the past indicative).
Properties of correct answer (A): Agr:yes, Number:Pl, Gender:F, attractor:yes, TensePremise:IFutAnt, TenseConsequent:IFut.
Violations in answer set:
B - attractor:no, Agr:Pl;
C - Number:Sg, Gender:M, tense:Cpas, attractor:no;
D - Gender:M, attractor:no;
E - Agr:no, Number:Sg, tense:CtrPas;
F - attractor:no.

## 5 Can ChatGPT solve BLMs?

The initial motivation to create this kind of task is to study how to guide current state of the art systems towards better generalisation. Given the current very high performance of large language models, however, it needs to be established first if current state of the art systems are already capable of solving this kind of task, or if it is challenging for them, and more importantly by which linguistic and reasoning processes they proceed to find a solution. In particular, the complex mixture of grammatical and abductive reasoning that is involved in BLMs yields an interesting test set to see if current LLM are indeed good at the formal part of language (syntax and semantics), but weak on the functional part (pragmatics and reasoning), as has been recently argued (Mahowald et al., 2023).

To get a general sense if ChatGPT solves the problem in the same way as human subjects solve RPMs, the instructions we give follow the same protocols as those given to subjects in Carpenter et al. (1990) (see Figure 4 and Figure 5). We perform two tests. First, we use the publicly available ChatGPT 3.5, in single one-day sessions, a different day for each type of matrix, to collect qualitative interactions. Then we use ChatGPT-4, over three days, for a more systematic collection of quantitative results. When appropriate, we give the sentences in Italian to reduce the possibility that ChatGPT has learned the structure of the problem from papers previously published on the Internet.

We will reach the conclusion that ChatGPT does not really manage to distill the underlying rules of the problem, but this should not detract from the first striking observation: it is entirely possible to have a series of interactive exchanges with ChatGPT on this task in a completely coherent way. This impressive result confirms the formal ability of current LLMs to process language. However, the analysis of the interactions shows that ChatGPT3.5 struggles with the rule-based nature of the task, confirming that its functional ability is still relatively limited. Even in its simplest form and with a lot of guidance from the prompts and dialogue, the nature of the problem is not understood and reliable answers are not reached. Here we give a summary of some interactions. The detailed protocols are in the appendix. As will be shown in the next section, ChatGPT4 exhibits better performance, as observed in many other cases, but the overall result is confirmed: it does not manage to see the underlying rules.

### 5.1 Subject-verb Agreement

Interaction 1, 2 and 3 (reported in the appendix) show that ChatGPT3.5 correctly identifies the three objects (subject, first attractor, second attractor) after inspecting the context, and correctly identifies their attributes (singular, plural) after inspecting the answers. Somewhat incorrectly, it comments on the semantic coherence of the sentence. It finds the right answer, but it eliminates alternatives for incorrect reasons. ChatGPT3.5 is not really finding the 'sequence' notion of the context but is mapping each answer sentence to a set of objects and attributes, that it appears to have established as necessary. The example matrix used is the one in

Figure 4: Prompt instructions for interaction 3 of the subject-verb agreement problem with ChatGPT 3.5. The response and other interactions are shown in the appendix. See text for a discussion.

Figure 5: Prompt instructions for interaction 1 of the causative alternation problem with ChatGPT 3.5. The response and other interactions are shown in the appendix. See text for a discussion.

Figure 4. In interaction 4, we explicitly tell it to look for grammatical patterns. The answer shows that ChatGPT3.5 can produce a coarse syntactic analysis. In interactions 5 to 8, we give more detail about number agreement patterns, and that it needs to look for patterns across sentences.

Interaction 9 is very interesting as, at this point, we are basically giving the problem away, but still ChatGPT3.5 cannot solve it. The answer shows that it parses number agreement incorrectly, so it assigns the wrong number to the nouns phrases and consequently cannot see the pattern. So, in interaction 10, we go back to Italian, where number agreement is more clearly marked on the verb and the noun phrases. The following interactions 10 to 19 clearly show that even with a lot of prompting ChatGPT3.5 is incapable of finding the global alternation pattern. It seems that ChatGPT3.5 never reaches the right combination of accuracy in identifying number agreement patterns, and accuracy in seeing patterns across the sentences.

## 5.2 Causative

The interactions about the causatives are impressive, despite the fact that a correct solution is not found. This test is both harder and easier than the previous one. The structure of the matrix is easier, as the sequence of stimuli is organised into two analogous subsequences, as can be easily seen in Figure 5. So the solution to this puzzle requires figuring out the objects, their purpose and their sequence of the first four sentences and apply it to the

sentences 5-8, to find solution A. However, ChatGPT does not see the pattern, both in interaction 1, where no instruction is given and interaction 2, where instruction is given about the fact that the sentences should be looked at as two block. The individual sentence by sentence explanations of what the sentences in the context are supposed to show is however very impressive (and correct). This is shown in Figure 6. To verify that this apaprent insightfulness does not come from simply repeating, parrot-like, snippets of the large literature on this problem in English, we repeat the interaction Italian, where this problem is much less studied. The Italian interaction repeats exactly the English pattern of reasoning (making us think some translation is going on internally) and no correct solution is found.

## 5.3 Sequence of Tenses and Mixture of Problems

An example of the prompt for the sequence of tenses problem can be seen in the appendix. An example of the prompt for the mixture of problems agreement and sequence of tenses is shown in Figure 7. We show interaction 1. As can be seen, we specify explicitly that the solution is not to be found in the logical sequence and the topics of the sentences.

The responses from ChatGPT3.5 for the sequence of tenses problem, which can be inspected in the appendix, show that while the explanations of the selection of the correct answer are not very con-

To solve the BLM-CausE problem, we need to analyze the given context sentences and the set of answer sentences to determine the most appropriate continuation for the context. Let's go through the process step by step:

Identify the relevant elements: In this context, we have two main elements: "the door" and "the butter." We need to observe the causality between these elements and their states. Analyze the context:

Sentence 1: The boy opened the door voluntarily. This sentence establishes that the boy was responsible for opening the door.

Sentence 2: The door opened suddenly. This sentence suggests that the door opened on its own accord, without external agency.

Sentence 3: The door was closed. This sentence indicates the initial state of the door, which was closed.

Sentence 4: The door is open. This sentence reveals the current state of the door, which is open.

Sentence 5: The cook melted the butter voluntarily. This sentence states that the cook caused the butter to melt.

Sentence 6: The butter melted slowly. This sentence describes the process of the butter melting.

Sentence 7: The butter was hard. This sentence states the initial state of the butter, which was hard.

Consider the answer choices: Now let's analyze each answer choice in relation to the context:

Answer A: The butter is melted. This answer suggests that the current state of the butter is melted. However, it doesn't directly relate to the given context. We don't have any information about the current state of the butter.

Answer B: The boy is open. This answer doesn't make logical sense as it equates the boy with being open, which is unrelated to the context.

Answer C: The door was open. This answer corresponds to the progression of the door's state as mentioned in the context. It follows the pattern established in sentences 2 and 3, where the door opens suddenly after being closed. This seems to be a reasonable continuation.

Answer D: The cook is melted. This answer equates the cook with being melted, which doesn't make sense in the context.

Answer E: The butter was melted. This answer corresponds to the progression of the butter's state as mentioned in the context. It follows the pattern established in sentences 5 and 6, where the cook causes the butter to melt slowly. This also seems like a plausible continuation.

Final decision: Comparing answers C and E, both seem to be reasonable continuations based on the given context. However, since the initial prompt requests choosing only one answer, we need to make a decision. Considering the sequential pattern of events in the context, it seems more likely that the door's state would be the next relevant topic. Therefore, the correct answer is C: The door was open.

Figure 6: Reply to interaction 1 of the causative alternation problem with ChatGPT 3.5. See text for a discussion.

I am going to ask you to solve a grammatical problem called BLM-MixI. I will give you a list of Italian sentences (numbered from 1 to 7), that I will call the context, and I will also give you a set of Italian sentences (identified by capital letters) that I will call the answers. I will ask you to choose among the answer sentences the one sentence that could be the next sentence in the context list of sentences. The sentences have been generated in a rule-based manner.
I am asking you to give me the correct answer, but also to describe step by step the hypotheses you are entertaining while you are solving the problem. I am also giving you a hint: This is a grammatical problem, not a logical problem. The solution does not lie in finding a logical continuation, but a continuation that follows the underlying grammatical patterns.

Context
1 Se fossi ricco, allora sarei felice.
2 Se fossero buoni, allora non sarebbero ricchi.
3 Se fossi stata ricca, allora non saresti stata felice.
4 Se foste state buone, allora sareste state ricche.
5 Se sarai bravo, indipendentemente dalle circostanze, allora sarai ricco.
6 Se sarete stati bravi, nonostante tutto, allora sarete ricchi.
7 Se sarai buona, tenendo conto di tanti altri fattori, allora sarai ricca.

Answers
A Se sarete state buone, e con un po' di fortuna, allora sarete ricche.
B Se sarai buona, allora sarai felice.
C Se fossi stato buono, allora fossi felice.
D Se sarete stati buoni, allora sarete ricchi.
E Se fossi stata buona, tenendo conto di tanti altri fattori, allora fossero stati felici.
F Se sarete state buone, allora sarete ricche.

Figure 7: Prompt instructions for interaction 1 of the mixture of agreement and tense sequence problems with ChatGPT 3.5. The response and other interactions are shown in the appendix. See text for a discussion.

vincing, when explicitly asked to formulate the rule, ChatGPT correctly identifies it as the sequence of tense rule. It insists, however, in finding semantic sequentiality, which is, instead, not relevant.

For the mixture of the two problems, agreement and sequence of tenses, we see that ChatGPT3.5 cannot even converge on a single answer. This is because, while it understands reasonably well the relevance of the sequence of tenses, it misses the agreement pattern completely, and so it cannot eliminate many incorrect alternative answers that depend on recognising this pattern.

## 5.4 Discussion

If we map impressionistically ChatGPT's solutions to the three areas in which people find difficulty (findings correspondences, goal and subgoal management and item novelty), we notice that ChatGPT is quite good at correspondence finding, but only if the problem encodes a single grammatical rule. Otherwise, it does not manage to identify the two patterns in the mixture matrix. Also, it clearly performs better with lexical semantic problems than morphological problems. With quite a bit of guidance, it seems to be able to identify an abstract template from lexically varied items (item novelty). However, it never manages to identify the sequence pattern, and consequently find the answer in any of the problems: this seems to be indicative of lack of goal and subgoal management.

## 6 Quantitative Analysis

The interaction illustrated in the previous section are a limited sample. We also developed a controlled sample of interactions, and analyse it quantitatively. We use ChatGPT-4 (between the 28th and the 30th of September, 2023).

### 6.1 Sample creation and quantitative measures

The dimensions of variation of the sample reflect the properties we need to analyse. We test two simple BLMs (Caus, Agr) and the mixed BLMs that merges Agreement and sequence of tense patterns (Mix). Each type of BLM has two versions, exhibiting two levels of lexicalisation (what in other work on BLMs has been called type I and a type III). Type I lexicalisation uses the same lexical items across all sentences in the matrix (see for example Figure 7), while the type III has complete lexical variability (see for example Figure 4).

For each combination of BLM simple problem and lexicalisation level we create five instances, built semi-automatically by transformation of a prototype with ChatGPT4 and then corrected or modified. Mixed BLMs where composed by hand. Some examples are shown in Appendix E.1.

The coding protocol is developed for comparability with the quantitative analyses developed for human subjects (Carpenter et al., 1990). We follow the same dimensions of evaluation (also discussed in the previous section): correspondence finding (elements and attributes), goal and subgoal management (timing and organisation of mentions, method used to find answers) and item novelty (type I and type III levels of lexical variation). The exact coding scheme corresponding to these concepts varies for each BLM problem, and is shown in Figure 8.

### 6.2 Quantitative results

The output were then manually analysed according to the categories shown in Figure 8. In Table 1, we report the summaries of the results and related statistics. The complete results tables are shown in Appendix E. The complete outputs for all problems are provided at `https://github.com/CLCL-Geneva/BLM-ChatGPTInteractions`.

**Success rate** The success rate is quite good, with a high rate of correctly identified answers. Nonetheless, the answers that are correct because the right pattern has been identified are fewer, with the ex-

| CORRECTLY IDENTIFIED ELEMENTS | |
| --- | --- |
| Caus | Subj Trans1, Obj Trans1, Subj Trans2, Obj Trans2 |
| Agr | NP1, NP2, NP3, V |
| SoT | Premise, Consequent |

| CORRECTLY IDENTIFIED ATTRIBUTES | |
| --- | --- |
| Caus | Volition, Ext. Agent, CoS:Initial, CoS:Current Agent, Patient |
| Agr | Sing, Plural, Match,Mismatch |
| SoT | Tense, Person, Gender, Match, Mismatch |

| GOAL MANAGEMENT | |
| --- | --- |
| All | One-at-a-Time, Pairwise, Row/Col-wise, Unordered |

| FINDING ANSWERS | |
| --- | --- |
| All | Rule-based generation, Answer elimination |

Figure 8: Elements and attributes and problem-solving procedures. Mixed problems are the union of the elements and attributes and intersection of problem-solving procedures.

treme case being the causative alternation, which has been identified as such only in one case.[1]

**Correspondence finding** The number of correctly identified elements is relatively high, ranging from 73 to 89 percent, confirming that ChatGPT4 (like 3.5) is good at correspondence finding: identifying the aspect of the problem it needs to pay attention to. Comparing the ranking, best for Agr and worse for Mix, to the previous table of scores of correct answers, it can be noticed that good identification of the objects does not guarantee proportionally good finding of the correct answer.

The percentage of correctly identified attributes is lower, showing that this aspect of the problem is harder to identify, and particularly hard for causative BLMs. This is actually not surprising, as the attributes of Caus-BLM are more abstract, while Agr and Mix are much visible morphologically. This aspect of correspondence finding is a better predictor of overall performance.

**Item novelty** The sensitivity to item novelty results from comparisons of BLMs of different levels of lexicalisation (type I, type III). The sensitivity to more or fewer new lexical items seems to vary by problem. It appears to be, in general, true that more variation in lexicalisation creates difficulty.

---

[1]Notice that, in coding answers, we gave a full credit for only partial identification of pattern. For example, we have coded correct identification of the agreement pattern in most cases when in fact only the subject alternation pattern was identified but the two-step patterns for the second NP was not identified (which was in most cases).

| SUCCESS RATE | | | |
|---|---|---|---|
| | Correct answers | Correct patterns | Total answers |
| Caus | 11 | 1 | 11 |
| Agr | 8 | 9 | 12 |
| Mix | 9 | 6 | 10 |

| CORRESPONDENCE FINDING | | | |
|---|---|---|---|
| | Identified elements (%) | Tot | Identified attributes(%) | Tot |
| Caus | 56 (85) | 66 | 20 (30) | 66 |
| Agr | 43 (89) | 48 | 42 (87) | 48 |
| Mix | 44 (73) | 60 | 72 (80) | 90 |
| Tot | 143 | 174 | 134 | 204 |

| GOAL MANAGEMENT | | | | | | |
|---|---|---|---|---|---|---|
| | Organisation of Mentions | | | | Finding Answers | | Tot |
| | OaT | Prw | RCw | Un | RbG | Ael | |
| Caus | 1 | 10 | 10 | 0 | 10 | 11 | 11 |
| Agr | 9 | 3 | 4 | 2 | 11 | 12 | 12 |
| Mix | 5 | 5 | 10 | 0 | 6 | 9 | 10 |
| Tot | 15 | 18 | 24 | 2 | 27 | 32 | |

Table 1: Quantitative results of BLM solutions (Id'ed els: identified elements; Id'ed attrs: identified attributes; OaT: One-at-a-Time; Prw:Pairwise; RCw: Row/Col-wise; Un: Unordered; RbG: Rule-b generation; Ael: Answer elimination).

More specifically, causatives BLM are quite different in type I and type III because the elements that alternate are no longer the same NPs. Yet, the results are very similar, so we do not notice a difference in complexity. Agreement BLMs show correct answer for Type I in all 6 cases, but only 2 for type III. So lexicalisation does indeed make the task harder. Also in the Mixed BLMs, type III is harder.

**Goal Management** The quantification of aspects of the reasoning employed to solve the problem show that the categories developed for humans apply easily here too, as the output of ChatGPT4 is coherent, articulate and corresponds well to the expectations. Items are inspected comparatively more often than one at a time and, depending on the problem, what are formulated as attempts to find patterns gives rise to pairwise rules. The inspection of sentences is row by row or by groups of rows and rarely unordered. Of the two methods of rule finding documented in humans, rule-based hypothesis followed by matching with answers and elimination of the incorrect ones is the most common. Only in a few cases, in the clearly harder mixed BLMs, no rules were hypothesised and only directly answer elimination was attempted.

# 7 Further Related Work

Traditionally, progressive matrices as intelligence tests are designed by hand, but recent research in vision that has used this task to train neural networks has typically employed some structured generative model to create a large number of questions. Wang and Su (2015) generate RPMs using first-order logic, which have further inspired Barrett et al. (2018a) to use relation-object-attribute triple instantiations. Zhang et al. (2019) use the Attribute Stochastic Image Grammar (A-SIG, proposed by Zhu and Mumford, 2006) as the representation of RPMs, and have created the Relational and Analogical Visual rEasoNing (RAVEN) dataset. The controlled construction of the context data and of the contrastive answer set is crucial. In this way, the learning process basically discovers how to induce the model and it makes it possible to identify clear dimensions of successful and unsuccessful generalisation. For example, matrices for vision have shown that the best models can apply known abstract relationships in novel combinations, but fail in applying known abstract relationships to unfamiliar entities (Barrett et al., 2018b).

# 8 A Roadmap for the Future

The investigation reported here aims to prove a concept: curated data and in-depth qualitative analyses inspired by human intelligence tests can complement large-scale benchmarking in an interesting way. In particular, large-scale benchmarking based on correlations of humans and machines does not enlighten us on the causal sources of behaviours, and instead hypothesis testing experiments need to be performed. It is also interesting to collect data about human and machine overt knowledge, rather than phychophysical measures (eye-tracking, brain activations) or internal ANN layer representations.

A larger-scale investigation applying the curated data and qualitative method presented here is however also possible. We need to develop multiple datasets and run several interactive experiments that need then to be measured and assessed by proper statistical methods.

# 9 Acknowledgements

We gratefully acknowledge the partial support of this work by the Swiss National Science Foundation, through grants #51NF40_180888 (NCCR Evolving Language) and SNF Advanced grant TMAG-1_209426 to PM.

# 10 Ethics Statement

To the best of our knowledge, there are no ethics concerns with this paper.

# 11 Limitations

Our preliminary conclusions about human-machine correspondence sketch the avenue for future developments. The work presented here is meant as a programmatic call. The main limitations concern limitations in the BLM development, in the range of examined architectures and the level of internal analysis, and in the range of cognitive concepts investigated. BLMs are being developed with larger-scale multi-linguality and greater variety of linguistic phenomena. Work is also planned to study how they affect internal layers of LLMs.

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

# A Agreement

N.B. The interactions are formatted for readability. In the prompt, we provide the context and answer examples only for those interactions where they change with respect to the previous interaction. ChatGPT repeats the context and answer sentences literally at every interaction. For readability we have introduced ellipses, indicated with (...).

## A.1 Interaction 1

I am going to ask you to solve a puzzle called BLM-AgrI. The language of the experiment is Italian. I will give you a list of sentences, that I will call the context, and I will also give you a set of sentences that I will call the answers. I will ask you to choose among the answer sentences the sentence that could be the next sentence in the context list of sentences. I am asking you to give me the correct answer in Italian but also to describe in English step by step the hypotheses you are entertaining while you are solving the problem.

Context
        Il computer col programma non funziona.
        I computer  col programma non funzionano.
        Il computer coi programmi  non funziona.
        I computer  coi programmi  non funzionano.
        Il computer col programma dell'esperimento non funziona.
        I computer  col programma dell'esperimento non funzionano.
        Il computer coi programmi  dell'esperimento non funziona.

Answers
        Il computer col programma e l'esperimento non funziona.
        I computer  coi programmi  dell' esperimento non funzionano.
        Il computer col programma non funziona.
        Il computer coi programmi  dell' esperimento non funzionano.
        I computer  col programma dell' esperimento non funziona.
        I computer  coi programmi  degli esperimenti  non funzionano.

---

ChatGPT

To solve the puzzle, I will analyze the given context and the set of answer sentences to identify the sentence that could be the next one in the context. Here's the step-by-step thought process:

I start by examining the context sentences: (...) I observe that the context consistently mentions computer(s) and program(s), along with their functionality (funzionare or non funzionare). The addition of "dell'esperimento" indicates that the computers and programs are related to an experiment.

Moving on to the answer sentences: (...) Comparing the answer sentences with the context, I need to find a sentence that fits logically and syntactically. It should mention the computer(s), program(s), and the experiment.

The answer sentence "Il computer col programma e l'esperimento non funziona" doesn't match the context because it uses the singular form "funziona" instead of the plural form "funzionano."

The answer sentence "I computer coi programmi dell'esperimento non funzionano" is consistent with the context. It maintains the plural form and correctly references the experiment.

The answer sentence "Il computer col programma non funziona" matches the previous context sentence but doesn't include the experiment aspect.

The answer sentence "Il computer coi programmi dell'esperimento non funzionano" aligns with the context by using the plural form and mentioning the experiment.

The answer sentence "I computer col programma dell'esperimento non funziona" has a disagreement in number. "I computer" is plural, while "non funziona" is singular.

The answer sentence "I computer coi programmi degli esperimenti non funzionano" incorrectly uses the singular form "non funziona" instead of the plural form "non funzionano."

Based on this analysis, the answer sentence that could be the next in the context is "I computer coi programmi dell'esperimento non funzionano."

Correct answer in Italian: "I computer coi programmi dell'esperimento non funzionano."

## A.2 Interaction 2

This interaction is a minimal variant of interaction 1 to see if ChatGPT understands that attractors, the noun phrases intervening between the subject and the verb, can also be decreasing in number, depending on the structure of the matrix.

I am going to ask you to solve a puzzle called BLM-AgrI.
The language of the experiment is Italian.
I will give you a list of sentences, that I will call the context, and I will also give you a set of sentences that I will call the answers.
I will ask you to choose among the answer sentences the sentence that could be the next sentence in the context list of sentences.
I am asking you to give me the correct answer in Italian but also to describe in English step by step the hypotheses you are entertaining while you are solving the problem.

Context
     Il computer col programma dell'esperimento non funziona.
     I computer col programma dell'esperimento non funzionano.
     Il computer coi programmi dell'esperimento non funziona.
     I computer coi programmi dell' esperimento non funzionano.
     Il computer col programma non funziona.
     I computer col programma non funzionano.
     Il computer coi programmi non funziona.

Answers
     Il computer e i programmi non funzionano.
     I computer coi programmi non funzionano.
     Il computer col programma non funziona.
     Il computer coi programmi dell'esperimento non funzionano.
     I computer col programma dell'esperimento non funziona.
     I computer coi programmi degli esperimenti non funzionano.

ChatGPT
To solve the puzzle, I will analyze the given context and the set of answer sentences to identify the sentence that could be the next one in the context. Here's the step-by-step thought process:
I start by examining the context sentences: (...)
I observe that the context consistently mentions computer(s) and program(s), along with their functionality (funzionare or non funzionare). The addition of "dell'esperimento" indicates that the computers and programs are related to an experiment.
Moving on to the answer sentences: (...)
Comparing the answer sentences with the context, I need to find a sentence that fits logically and syntactically. It should mention the computer(s), program(s), and the experiment.
The answer sentence "I computer e i programmi non funzionano" is grammatically correct and maintains the plural form for both "computer" and "programmi."
The answer sentence "I computer coi programmi non funzionano" matches the context by using the correct form and maintaining the plural form for both "computer" and "programmi."
The answer sentence "Il computer col programma non funzionano" has a disagreement in number. "Il computer" is singular, while "non funzionano" is plural.
The answer sentence "Il computer coi programmi dell'esperimento non funzionano" is consistent with the context. It maintains the plural form and correctly references the experiment.
The answer sentence "I computer col programma dell'esperimento non funziona" has a disagreement in number. "I computer" is plural, while "non funziona" is singular.
The answer sentence "I computer coi programmi degli esperimenti non funzionano" incorrectly uses the plural form "degli esperimenti" instead of the singular form "dell'esperimento" mentioned in the context.
Based on this analysis, the answer sentence that could be the next in the context is "I computer e i programmi non funzionano."
Correct answer in Italian: "I computer e i programmi non funzionano."

## A.3 Interaction 3

We vary the words to see if ChatGPT can group objects and attributes in a more abstract way. Here, we vary all words in each sentence (type III data). We also number the context sentences to highlight the notion of list. We also change the language from Italian to English.

I am going to ask you to solve a puzzle called BLM-AgrE. I will give you a list of sentences (numbered from 1 to 7), that I will call the context, and I will also give you a set of sentences (identified by capital letters) that I will call the answers. I will ask you to choose among the answer sentences the sentence that could be the next sentence in the context list of sentences.
I am asking you to give me the correct answer, but also to describe step by step the hypotheses you are entertaining while you are solving the problem.
Context
1 The talk on history has started later than expected.
2 Those responsible for the right will resign.
3 The show with the paintings has met with great success.
4 The threats of reforms worry the doctors.
5 The bunch of keys of the cell sits on the shelf.
6 The studies on the effect of the drug will appear soon.
7 The threat of reforms in the school worries the doctors.
Answers
A The tablecloths on the table and the console shine in the sun.
B The friends of the owners of the villa were sleeping on the beach.
C The lawyers of the murderers will come back.
D The lawyers of the murderers of the village will come back.
E The visit of the palace of the crafts is approaching.
F The computers with the program of the experiment are broken.

ChatGPT
To solve the puzzle, I will analyze the given context sentences and the set of answer sentences to identify the sentence that could be the next one in the context. Here's the step-by-step thought process:
I start by examining the context sentences: (...)
I observe that the context sentences are unrelated to each other, discussing various topics such as a talk on history, responsible individuals resigning, a successful art show, doctors' concerns about reforms, keys on a shelf, drug studies, and the threat of reforms in the school.
Moving on to the answer sentences: (...)
Considering the context, I look for an answer sentence that fits logically and thematically with the preceding sentences.
The answer sentence "The tablecloths on the table and the console shine in the sun" does not relate to any of the previous context sentences or their topics.
The answer sentence "The friends of the owners of the villa were sleeping on the beach" is unrelated to the context and introduces a new scenario.
The answer sentence "The lawyers of the murderers will come back" is unrelated to the context and introduces a criminal element not previously mentioned.
The answer sentence "The lawyers of the murderers of the village will come back" introduces a new scenario involving murderers and a village, which is not relevant to the context.
The answer sentence "The visit of the palace of the crafts is approaching" is unrelated to the context and introduces a visit to a palace, which has not been mentioned before.
The answer sentence "The computers with the program of the experiment are broken" is thematically consistent with the previous context sentence mentioning the studies on the effect of a drug and the experiments. It introduces the concept of broken computers, which could be a logical continuation.
Based on this analysis, the answer sentence that could be the next in the context is "The computers with the program of the experiment are broken."
Correct answer: F) The computers with the program of the experiment are broken.

## A.4 Interaction 4

Like interaction 3, we vary all the words to see if ChatGPT can group objects and attributes in a more abstract way. However, since the previous interactions have shown that ChatGPT deals with the sentences based on topics and network of concepts and tries to solve the puzzle as a logical or semantic problem, we give it more information on the fact that it needs to focus on the syntactic form of the sentences. So the instruction is slightly different, we no longer talk of *puzzle*, but of *grammatical problem*. The context and answer sentences are the same as those in interaction 3.

I am going to ask you to solve a grammatical problem called BLM-AgrE.
I will give you a list of sentences (numbered from 1 to 7) , that I will call the context, and I will also give you a set of sentences (identified by capital letters) that I will call the answers.
I will ask you to choose among the answer sentences the sentence that could be the next sentence in the context list of sentences.
I am asking you to give me the correct answer, but also to describe step by step the hypotheses you are entertaining while you are solving the problem.

ChatGPT
To solve the grammatical problem, I will analyze the given context sentences and the set of answer sentences to identify the sentence that could be the next one in the context. Here's the step-by-step thought process:
I start by examining the context sentences: (...)
I observe that the context sentences are unrelated to each other, discussing various topics such as a talk on history, responsible individuals resigning, a successful art show, doctors' concerns about reforms, keys on a shelf, drug studies, and the threat of reforms in the school.
Moving on to the answer sentences: (...)
Considering the grammatical structure of the context, I look for an answer sentence that could continue the pattern of "The [noun phrase] [verb phrase]."
The answer sentence "The tablecloths on the table and the console shine in the sun" does not fit the grammatical pattern of the context sentences.
The answer sentence "The friends of the owners of the villa were sleeping on the beach" does not follow the grammatical structure of the context.
The answer sentence "The lawyers of the murderers will come back" does not match the grammatical pattern of the context.
The answer sentence "The lawyers of the murderers of the village will come back" does not adhere to the grammatical structure of the context.
The answer sentence "The visit of the palace of the crafts is approaching" does not conform to the grammatical pattern of the context.
The answer sentence "The computers with the program of the experiment are broken" matches the grammatical structure of the context, which consists of "The [noun phrase] [verb phrase]." It introduces the concept of broken computers, which could be a logical continuation.
Based on this analysis, the answer sentence that could be the next in the context is "The computers with the program of the experiment are broken."
Correct answer: F) The computers with the program of the experiment are broken.

## A.5 Interaction 5

This is a continuation of interaction 4. We still no longer talk of puzzle but of grammar rule and try to give it more hints.

> The answer is not correct but you are on the good path, can you try to solve the problem with a more complex structure for the context sentences?

> ChatGPT
>
> Apologies for the previous incorrect answer. Let's approach the problem with a more complex structure for the context sentences. Here's a revised step-by-step thought process:
> Re-examining the context sentences with a more complex structure: (...)
> Analyzing the context sentences, I observe that they all follow a pattern of "The [noun phrase] [verb phrase] [adverbial phrase]."
> Moving on to the answer sentences: (...)
> Considering the revised grammatical structure of the context, I look for an answer sentence that adheres to the pattern of "The [noun phrase] [verb phrase] [adverbial phrase]."
> The answer sentence "The tablecloths on the table and the console shine in the sun" matches the grammatical structure of the context and introduces the idea of shining in the sun, which can be seen as a continuation.
> The answer sentence "The friends of the owners of the villa were sleeping on the beach" does not align with the grammatical structure of the context.
> The answer sentence "The lawyers of the murderers will come back" does not match the grammatical structure of the context.
> The answer sentence "The lawyers of the murderers of the village will come back" does not conform to the grammatical structure of the context.
> The answer sentence "The visit of the palace of the crafts is approaching" does not adhere to the grammatical pattern of the context.
> The answer sentence "The computers with the program of the experiment are broken" does not follow the revised grammatical structure of the context.
> Based on this analysis, the answer sentence that could be the next in the context, considering the revised grammatical structure, is "The tablecloths on the table and the console shine in the sun."
> Correct answer: A) The tablecloths on the table and the console shine in the sun.

## A.6 Interaction 6

> ok, not quite there yet, let me tell you and see if you can discover the correct rules. The property you need to pay attention to is number agreement.

> ChatGPT
>
> Thank you for the additional clarification. Let's consider number agreement as a key property to find the correct answer. Here's the revised step-by-step thought process:
> Revisiting the context sentences: (...)
> Analyzing the context sentences, I notice that they all use the definite article "The" followed by a noun phrase and a verb phrase.
> Moving on to the answer sentences: (...)
> Considering the property of number agreement, I need to ensure that the noun phrase in the answer sentence matches the number (singular or plural) used in the context sentences.
> The answer sentence "The tablecloths on the table and the console shine in the sun" does not match the number agreement with the singular noun phrases used in the context sentences. The noun phrase "tablecloths on the table and the console" is plural, while the context sentences use singular noun phrases.
> The answer sentence "The friends of the owners of the villa were sleeping on the beach" also does not match the number agreement with the singular noun phrases used in the context sentences. The verb "were" indicates a plural subject, but the context sentences use singular subjects.
> The answer sentence "The lawyers of the murderers will come back" does not match the number agreement with the singular noun phrases used in the context sentences. The noun phrase "lawyers of the murderers" is plural, while the context sentences use singular subjects.
> The answer sentence "The lawyers of the murderers of the village will come back" also does not match the number agreement with the singular noun phrases used in the context sentences. The noun phrase "lawyers of the murderers of the village" is plural, while the context sentences use singular subjects.
> The answer sentence "The visit of the palace of the crafts is approaching" matches the number agreement with the singular noun phrases used in the context sentences. The noun phrase "visit of the palace of the crafts" is singular, matching the context sentences.
> The answer sentence "The computers with the program of the experiment are broken" matches the number agreement with the plural noun phrases used in the context sentences. The noun phrase "computers with the program of the experiment" is plural, matching the context sentences.
> Based on this analysis, considering the property of number agreement, the answer sentence that fits the context is "The computers with the program of the experiment are broken."
> Correct answer: F) The computers with the program of the experiment are broken.

## A.7 Interaction 7

Not quite, you need to find the continuing sentence based on a pattern of number agreement.

ChatGPT

Apologies for the oversight. Let's revisit the context and analyze the sentences while paying attention to number agreement as the determining factor for finding the correct answer. Here's the revised step-by-step thought process:

Revisiting the context sentences: (...)

Analyzing the context sentences, I notice that the noun phrases in each sentence agree in number with the corresponding verb phrases. In other words, singular noun phrases are paired with singular verbs, and plural noun phrases are paired with plural verbs.

Moving on to the answer sentences: (...)

Considering the property of number agreement, I need to find an answer sentence where the noun phrase agrees in number with the verb phrase, similar to the pattern seen in the context sentences.

Answer sentences A, B, C, and D do not demonstrate number agreement with the context sentences.

Answer sentence E, "The visit of the palace of the crafts is approaching," maintains number agreement with the context sentences. The noun phrase "the visit of the palace of the crafts" is singular, matching the singular verb "is approaching" used in the context sentences.

Answer sentence F, "The computers with the program of the experiment are broken," also demonstrates number agreement with the context sentences. The noun phrase "the computers with the program of the experiment" is plural, matching the plural verb "are broken" used in the context sentences.

Comparing the two remaining answer sentences, both E and F demonstrate number agreement with the context sentences. However, based on the context and the potential continuation of the sentences, it is more likely that the next sentence would relate to the topic of the computers and the program.

Based on this analysis, considering the pattern of number agreement and the context, the answer sentence that best fits the continuation is:

Correct answer: F) The computers with the program of the experiment are broken.

## A.8 Interaction 8

Let me explain better. You need to find the continuing sentence based on a pattern of number agreement. The context gives you a pattern within each sentence but also across all sentences, so that the continuation sentence continues the global pattern of agreement as well as obeying the sentence-internal pattern of agreement.

ChatGPT

Thank you for the clarification. I apologize for the misunderstanding. Let's revisit the context and analyze the sentences while considering both the sentence-internal pattern of number agreement and the global pattern of agreement. Here's the revised step-by-step thought process:

Revisiting the context sentences: (...)

Analyzing the context sentences, I observe the following patterns of number agreement:

Sentences 1, 3, 5, and 6 have singular noun phrases that agree with singular verbs. Sentences 2, 4, and 7 have plural noun phrases that agree with plural verbs.

Moving on to the answer sentences: (...)

Considering both the sentence-internal pattern of number agreement and the global pattern observed in the context sentences, I need to find an answer sentence that continues the overall pattern of number agreement.

Answer sentences A, B, C, and D do not continue the global pattern of number agreement observed in the context sentences.

Answer sentence E, "The visit of the palace of the crafts is approaching," continues the sentence-internal pattern of number agreement by using a singular noun phrase and a singular verb. However, it does not continue the global pattern of plural noun phrases and plural verbs observed in the context sentences.

Answer sentence F, "The computers with the program of the experiment are broken," continues both the sentence-internal pattern of number agreement (plural noun phrase with plural verb) and the global pattern of plural noun phrases and plural verbs observed in the context sentences.

Considering the requirement to continue both the sentence-internal pattern and the global pattern of number agreement, answer sentence F aligns with the desired continuation.

Based on this analysis, the answer sentence that follows the pattern of number agreement both within each sentence and across all sentences is:

Correct answer: F) The computers with the program of the experiment are broken.

## A.9 Interaction 9

We give indication that a grammatical pattern is to be found without saying which one. We switch to Italian, where gramamtical number is more salient morphologically.

I am going to ask you to solve a grammatical problem called BLM-AgrI.
I will give you a list of sentences (numbered from 1 to 7) , that I will call the context, and I will also give you a set of sentences (identified by capital letters) that I will call the answers. I will ask you to choose among the answer sentences the sentence that could be the next sentence in the context list of sentences.
You need to find the continuing sentence based on a grammatical pattern. The context gives you a pattern within each sentence but also across all sentences, so that the continuation sentence continues the global pattern as well as obeying the sentence-internal pattern of agreement.
I am asking you to give me the correct answer, but also to describe step by step the hypotheses you are entertaining while you are solving the problem.
Context
1 La presentazione di storia comincia in ritardo.
2 I responsabili della legge daranno le dimissioni.
3 La mostra dei quadri ha incontrato grande successo.
4 Le minaccie di riforme preoccupano i dottori.
5 Il mazzo di chiavi della cantina pende dal gancio.
6 Gli studi sugli effetti del farmaco appariranno presto.
7 La frequenza di riforme nelle scuole preoccupa i dottori.
Answers
A Le tovaglie sul tavolo e la credenza brillano al sole.
B Gli amici dei proprietari della villa dormono sulla spiaggia.
C Gli avvocati degli assassini torneranno.
D Gli avvocati degli assassini del villaggio torneranno.
E La visita al palazzo delle arti si avvicina.
F I computer col programma dell'esperimento non funzionano.

Chat GPT
To solve the BLM-AgrI problem, I will carefully analyze the given context sentences and the provided answer options. I will consider both the sentence-internal pattern of agreement and the global pattern across all sentences. Here is the step-by-step thought process:
Revisiting the context sentences in Italian: (...)
Analyzing the context sentences, I observe the following patterns:
Sentences 1, 3, 5, and 6 have a singular noun phrase (La presentazione, La mostra, Il mazzo, Gli studi) paired with a singular verb phrase (comincia, ha incontrato, pende, appariranno).
Sentences 2, 4, and 7 have a plural noun phrase (I responsabili, Le minaccie, La frequenza) paired with a plural verb phrase (daranno, preoccupano).
Moving on to the answer sentences: (...)
Considering both the sentence-internal pattern of agreement and the global pattern observed in the context sentences, I need to find an answer sentence that continues both the sentence-internal and global patterns.
Answer sentences A, B, C, D, and E do not continue the global pattern observed in the context sentences.
Answer sentence F, "I computer col programma dell'esperimento non funzionano," continues the sentence-internal pattern of agreement (plural noun phrase with plural verb) observed in the context sentences. It also maintains the global pattern of plural noun phrases (I computer col programma dell'esperimento) and plural verbs (non funzionano).
Considering the requirement to continue both the sentence-internal and global patterns of agreement, answer sentence F aligns with the desired continuation.
Based on this analysis, the answer sentence that follows the grammatical pattern both within each sentence and across all sentences is:
Correct answer: F) I computer col programma dell'esperimento non funzionano.

## A.10 Interaction 10

Good job, but you made a mistake and then the conclusion was consequently wrong. If I tell you that 'Gli studi' is plural can you retry?

ChatGPT

Apologies for the mistake. Thank you for pointing out that "Gli studi" in sentence 6 is indeed plural. Let me revise my analysis based on that information:

Revisiting the context sentences in Italian: (...)

Analyzing the context sentences, I observe the following patterns:

Sentences 1, 3, 5, and 7 have a singular noun phrase paired with a singular verb phrase.

Sentences 2, 4, and 6 have a plural noun phrase paired with a plural verb phrase.

Moving on to the answer sentences: (...)

Considering the patterns observed in the context sentences and the requirement to continue both the sentence-internal and global patterns of agreement:

Answer sentences A, B, C, D, and E do not continue the global pattern of number agreement observed in the context sentences.

Answer sentence F, "I computer col programma dell'esperimento non funzionano," aligns with the sentence-internal pattern (plural noun phrase with plural verb) observed in the context sentences. However, it does not follow the global pattern of number agreement because all previous sentences have a singular subject paired with a singular verb phrase.

Therefore, considering the updated information, none of the answer sentences provided (A, B, C, D, E, F) continue the grammatical pattern both within each sentence and across all sentences.

Apologies for the confusion in the previous response.

## A.11 Interaction 11

Not quite, let me help you. The correct answer is B. Gli amici dei proprietari della villa dormono sulla spiaggia.

Keeping in mind the information I gave you before about sentence-internal patterns and global patterns, can you explain to me why?

ChatGPT

Apologies for the confusion. Let me reevaluate the context and analyze the patterns to explain why the correct answer is B. "Gli amici dei proprietari della villa dormono sulla spiaggia."

Revisiting the context sentences: (...)

Analyzing the patterns:

In sentences 1, 3, 5, and 7, we observe a singular noun phrase paired with a singular verb phrase. In sentences 2, 4, and 6, we observe a plural noun phrase paired with a plural verb phrase. Considering the answer sentences: (...)

The correct answer: Answer sentence B, "Gli amici dei proprietari della villa dormono sulla spiaggia," aligns with both the sentence-internal pattern and the global pattern of number agreement observed in the context sentences.

Sentence 2 in the context has a plural subject "I responsabili della legge" paired with a plural verb phrase "daranno le dimissioni." Answer sentence B follows the same pattern with a plural subject "Gli amici dei proprietari della villa" paired with a plural verb phrase "dormono sulla spiaggia."

Therefore, based on the patterns observed, sentence B continues the grammatical pattern within each sentence and maintains the global pattern of number agreement, making it the correct answer.

## A.12 Interaction 12

Can you tell me more explicitly what is the global pattern you have observed and why sentence B is the right continuation according to this pattern?

ChatGPT

Certainly! Upon further analysis, the global pattern observed in the context sentences is the agreement of the subject noun phrases with the corresponding verb phrases in terms of number (singular or plural).

The global pattern can be summarized as follows:

Sentences 1, 3, 5, and 7 have singular subject noun phrases paired with singular verb phrases.

Sentences 2, 4, and 6 have plural subject noun phrases paired with plural verb phrases.

In the given context, the sentence "Gli studi sugli effetti del farmaco appariranno presto" (Sentence 6) establishes the pattern of a plural subject ("Gli studi") paired with a plural verb phrase ("appariranno presto").

Now let's examine the provided answer sentences and see which one aligns with this global pattern.

Answer sentence B, "Gli amici dei proprietari della villa dormono sulla spiaggia," continues the global pattern. It consists of a plural subject noun phrase ("Gli amici dei proprietari della villa") paired with a plural verb phrase ("dormono sulla spiaggia"). This choice maintains the agreement of the subject noun phrase and the verb phrase, as seen in the context sentence 6, thus adhering to the global pattern observed.

Therefore, based on the global pattern of number agreement, sentence B is the correct continuation in the context list of sentences.

## A.13 Interaction 13

According to your reasoning also C and D should be correct because they also have plural subject matching plural verb. So why are they not the right answer?

ChatGPT
You're absolutely correct, and I apologize for the oversight. Based on the global pattern of number agreement, both answer sentences C and D, "Gli avvocati degli assassini torneranno" and "Gli avvocati degli assassini del villaggio torneranno," match the pattern of a plural subject ("Gli avvocati degli assassini") paired with a plural verb phrase ("torneranno"). Therefore, they also satisfy the global pattern observed in the context sentences.

As a result, considering the information provided, answer sentences B, C, and D could all be valid continuations based on the global pattern of number agreement. It seems there might not be a unique correct answer solely based on the global pattern.

To determine the specific correct answer, it would be helpful to consider additional linguistic factors or constraints that might be present in the puzzle. Without further information or specific instructions, it is difficult to definitively determine the correct answer solely based on the global pattern of number agreement.

## A.14 Interaction 14

There is only one correct answer and that is B, so you correctly say, C and D might be excluded by other patterns. Can you see what they are?

ChatGPT
Apologies for the confusion earlier. Upon closer examination, there are indeed other linguistic patterns present in the context sentences that help us exclude answer sentences C and D as the correct continuation. Let's analyze these patterns:

Sentence-Internal Pattern: Each sentence in the context exhibits a consistent sentence-internal pattern of number agreement between the subject noun phrase and the verb phrase. In sentences 1, 3, 5, and 7, we observe singular subject noun phrases paired with singular verb phrases. In sentences 2, 4, and 6, we see plural subject noun phrases paired with plural verb phrases.

Sentence-External Pattern: The context sentences also form a broader sentence-external pattern. We notice that the subject noun phrases in the context sentences refer to different entities or groups, each with their own contextually specific characteristics. For example, in sentence 1, "La presentazione di storia" refers to a singular event (the talk on history). In sentence 2, "I responsabili della legge" refers to a group of individuals responsible for the law.

Applying these patterns, we can see that answer sentences C and D, "Gli avvocati degli assassini torneranno" and "Gli avvocati degli assassini del villaggio torneranno," do not fit the broader sentence-external pattern established in the context. The subject noun phrases in these answer sentences refer to a specific group (the lawyers of the murderers or the lawyers of the murderers of the village). However, this specific group is not a consistent theme or contextually relevant entity throughout the previous context sentences. Therefore, based on the sentence-external pattern, answer sentences C and D can be excluded as the correct continuation.

In contrast, answer sentence B, "Gli amici dei proprietari della villa dormono sulla spiaggia," fits both the sentence-internal pattern of number agreement (plural subject paired with plural verb) and the broader sentence-external pattern. The subject noun phrase refers to a group of individuals (the friends of the owners of the villa), which aligns with the contextual theme of different entities being discussed in the previous sentences.

Considering the sentence-internal and sentence-external patterns, answer sentence B is the correct continuation that maintains consistency and relevance within the given context.

## A.15 Interaction 15

There is only one correct answer and that is B, so you correctly say, C and D might be excluded by other patterns. Focussing exclusively on patterns of agreement, can you see what they are?

ChatGPT

I apologize for the confusion earlier. Focusing exclusively on patterns of agreement, let's analyze the sentences again to identify the specific patterns at play:

La presentazione di storia comincia in ritardo. (Singular subject, singular verb)

I responsabili della legge daranno le dimissioni. (Plural subject, plural verb)

La mostra dei quadri ha incontrato grande successo. (Singular subject, singular verb)

Le minacce di riforme preoccupano i dottori. (Plural subject, plural verb)

Il mazzo di chiavi della cantina pende dal gancio. (Singular subject, singular verb)

Gli studi sugli effetti del farmaco appariranno presto. (Plural subject, plural verb)

La frequenza di riforme nelle scuole preoccupa i dottori. (Plural subject, plural verb)

Based on these patterns of agreement, we observe the following consistent rules:

Singular subjects are paired with singular verbs.

Plural subjects are paired with plural verbs.

Applying these patterns to the answer sentences, we find that answer B, "Gli amici dei proprietari della villa dormono sulla spiaggia," adheres to the pattern of agreement. The subject "Gli amici" (plural) is appropriately matched with the verb "dormono" (plural), maintaining the pattern observed in the context sentences.

In contrast, answer sentences C and D, "Gli avvocati degli assassini torneranno" and "Gli avvocati degli assassini del villaggio torneranno," do not conform to the pattern of agreement. Although they have plural subjects ("Gli avvocati"), the verb "torneranno" (plural) does not match with the singular noun phrases ("degli assassini" and "del villaggio").

Therefore, based solely on the patterns of agreement, answer sentence B is the correct continuation as it aligns with the consistent pattern of number agreement observed in the context sentences.

## A.16 Interaction 16

Both C and D follow the rule you mention so why are you saying they do not follwo the rule?

ChatGPT

Apologies for the confusion in my previous response. You are correct that both options C and D follow the pattern of plural subject-verb agreement. Therefore, they are potential valid answers based on the observed agreement rule.

To select the correct answer between C and D, we need to consider additional patterns or constraints within the context sentences. Since I do not have access to the specific global patterns or constraints you mentioned, I am unable to provide a definitive answer based solely on grammatical agreement.

If there are other patterns or constraints related to the context sentences that can help determine the correct answer, please provide further information, and I'll be glad to reassess and assist you accordingly.

## A.17 Interaction 17

The other constraints are also related to number and agreement, can you see the patterns that eliminate C and D?

ChatGPT

Apologies for the oversight. Let's reassess the context and the remaining options (C and D) considering other patterns related to number and agreement.

Context: (...)

Answers: C. Gli avvocati degli assassini torneranno.

D. Gli avvocati degli assassini del villaggio torneranno.

To identify the patterns that eliminate C and D, we need to examine the agreement between the subjects and the verbs in each option.

In option C, "Gli avvocati degli assassini torneranno" (The lawyers of the murderers will come back), the subject "Gli avvocati" (The lawyers) is plural, and the verb "torneranno" (will come back) agrees with the plural subject.

In option D, "Gli avvocati degli assassini del villaggio torneranno" (The lawyers of the murderers of the village will come back), the subject "Gli avvocati" (The lawyers) is plural, and the verb "torneranno" (will come back) also agrees with the plural subject.

Based on the information provided, both options C and D seem to follow the pattern of number agreement within the sentences. However, if there are additional patterns or constraints related to number and agreement that apply to the context sentences as a whole, please provide further details so that I can reassess and assist you more accurately.

# B  Causatives

N.B. The interactions are formatted for readability. In the prompt, we provide the context and answer examples only for those interactions where they change with respect to the previous interaction.

## B.1  Interaction 1

I am going to ask you to solve a grammatical problem called BLM-CausE. I will give you a list of English sentences (numbered from 1 to 7), that I will call the context, and I will also give you a set of English sentences (identified by capital letters) that I will call the answers. I will ask you to choose among the answer sentences the one sentence that could be the next sentence in the context list of sentences. The sentences have been generated in a rule-based manner.

I am asking you to give me the correct answer, but also to describe step by step the hypotheses you are entertaining while you are solving the problem.

Context
1 The boy opened the door voluntarily.
2 The door opened suddenly.
3 The door was closed.
4 The door is open.
5 The cook melted the butter voluntarily.
6 The butter melted slowly.
7 The butter was hard.

Answers
A The butter is melted.
B The boy is open.
C The door was open.
D The cook is melted.
E The butter was melted.

---

ChatGPT
To solve the BLM-CausE problem, we need to analyze the given context sentences and the set of answer sentences to determine the most appropriate continuation for the context. Let's go through the process step by step:
Identify the relevant elements:
In this context, we have two main elements: "the door" and "the butter." We need to observe the causality between these elements and their states.
Analyze the context:
Sentence 1: The boy opened the door voluntarily. This sentence establishes that the boy was responsible for opening the door.
Sentence 2: The door opened suddenly. This sentence suggests that the door opened on its own accord, without external agency.
Sentence 3: The door was closed. This sentence indicates the initial state of the door, which was closed.
Sentence 4: The door is open. This sentence reveals the current state of the door, which is open.
Sentence 5: The cook melted the butter voluntarily. This sentence states that the cook caused the butter to melt.
Sentence 6: The butter melted slowly. This sentence describes the process of the butter melting.
Sentence 7: The butter was hard. This sentence states the initial state of the butter, which was hard.
Consider the answer choices: Now let's analyze each answer choice in relation to the context:
Answer A: The butter is melted. This answer suggests that the current state of the butter is melted. However, it doesn't directly relate to the given context. We don't have any information about the current state of the butter.
Answer B: The boy is open. This answer doesn't make logical sense as it equates the boy with being open, which is unrelated to the context.
Answer C: The door was open. This answer corresponds to the progression of the door's state as mentioned in the context. It follows the pattern established in sentences 2 and 3, where the door opens suddenly after being closed. This seems to be a reasonable continuation.
Answer D: The cook is melted. This answer equates the cook with being melted, which doesn't make sense in the context.
Answer E: The butter was melted.
This answer corresponds to the progression of the butter's state as mentioned in the context. It follows the pattern established in sentences 5 and 6, where the cook causes the butter to melt slowly. This also seems like a plausible continuation.
Final decision: Comparing answers C and E, both seem to be reasonable continuations based on the given context. However, since the initial prompt requests choosing only one answer, we need to make a decision. Considering the sequential pattern of events in the context, it seems more likely that the door's state would be the next relevant topic. Therefore, the correct answer is C: The door was open.
In summary, the step-by-step analysis led us to select answer C: The door was open as the most appropriate continuation of the given context.

## B.2 Interaction 2

I am going to ask you to solve a grammatical problem called BLM-CausE. I will give you a list of English sentences (numbered from 1 to 7), that I will call the context, and I will also give you a set of English sentences (identified by capital letters) that I will call the answers. I will ask you to choose among the answer sentences the one sentence that could be the next sentence in the context list of sentences. The sentences have been generated in a rule-based manner and follow both internal rules and a global patterns that runs across all sentences, hence the notion of sequence.

I am asking you to give me the correct answer, but also to describe step by step the hypotheses you are entertaining while you are solving the problem.

To solve the BLM-CausE problem with the given context and answer choices, let's analyze the information step by step:

Identify the relevant elements: The relevant elements in this context are "the boy," "the door," "the cook," and "the butter."

Analyze the context:

Sentence 1: The boy opened the door voluntarily. This sentence establishes that the boy voluntarily performed the action of opening the door.

Sentence 2: The door opened suddenly. This sentence describes the door's action of opening, seemingly without external agency.

Sentence 3: The door was closed. This sentence provides the initial state of the door, which was closed.

Sentence 4: The door is open. This sentence indicates the current state of the door, which is open.

Sentence 5: The cook melted the butter voluntarily. This sentence states that the cook caused the butter to melt through their voluntary action.

Sentence 6: The butter melted slowly. This sentence describes the process of the butter melting, indicating the action happening over time.

Sentence 7: The butter was hard. This sentence represents the initial state of the butter, which was hard.

Consider the answer choices:

Let's analyze each answer choice in relation to the context:

Answer A: The butter is melted. This answer suggests that the current state of the butter is melted. However, it doesn't directly relate to the given context. We don't have any information about the current state of the butter.

Answer B: The boy is open. This answer doesn't make logical sense as it equates the boy with being open, which is unrelated to the context.

Answer C: The door was open. This answer corresponds to the logical progression of events in the context. It follows the pattern established in sentences 2 and 3, where the door opens suddenly after being closed. This seems to be a reasonable continuation.

Answer D: The cook is melted. This answer equates the cook with being melted, which doesn't make sense in the context.

Answer E: The butter was melted. This answer corresponds to the progression of events in the context. It follows the pattern established in sentences 5 and 6, where the cook causes the butter to melt slowly. This also seems like a plausible continuation.

Final decision: Comparing answers C and E, both could be reasonable continuations based on the given context. However, considering the global pattern and the notion of sequence mentioned in the problem description, it seems more likely that the next sentence would continue the discussion about the door's state. Therefore, the correct answer is C: The door was open.

In summary, by following the step-by-step analysis, we have determined that answer C: The door was open is the most appropriate continuation of the given context.

## B.3 Interaction 3

Here we stop the generation as it seems to be applying the same reasoning and could be using the translation.

I am going to ask you to solve a grammatical problem called BLM-CausI. I will give you a list of Italian sentences (numbered from 1 to 7) , that I will call the context, and I will also give you a set of Italian sentences (identified by capital letters) that I will call the answers. I will ask you to choose among the answer sentences the one sentence that could be the next sentence in the context list of sentences. The sentences have been generated in a rule-based manner and follow both internal rules and a global patterns that runs across all sentences, hence the notion of sequence.
I am asking you to give me the correct answer, but also to describe step by step the hypotheses you are entertaining while you are solving the problem.
Context
1 Il ragazzo apre la porta volontariamente.
2 La porta si apre di colpo.
3 La porta era chiusa.
4 La porta e' aperta.
5 Il cuoco scioglie il burro volontariamente.
6 Il burro si scioglie lentamente.
7 Il burro era duro.
Answers
A Il burro e' sciolto.
B Il ragazzo e' aperto.
C La porta era aperta.
D il cuoco e' sciolto.
E Il burro era sciolto.

To solve the BLM-CausI problem with the given context and answer choices in Italian, let's analyze the information step by step:
Identify the relevant elements: The relevant elements in this context are "il ragazzo" (the boy), "la porta" (the door), "il cuoco" (the cook), and "il burro" (the butter).
Analyze the context:
Sentence 1: Il ragazzo apre la porta volontariamente. (The boy opens the door voluntarily.) This sentence establishes that the boy voluntarily performs the action of opening the door.
Sentence 2: La porta si apre di colpo. (The door opens suddenly.) This sentence describes the door's action of opening, seemingly without external agency.
Sentence 3: La porta era chiusa. (The door was closed.) This sentence provides the initial state of the door, which was closed.
Sentence 4: La porta e' aperta. (The door is open.) This sentence indicates the current state of the door, which is open.
Sentence 5: Il cuoco scioglie il burro volontariamente. (The cook melts the butter voluntarily.) This sentence states that the cook causes the butter to melt through their voluntary action.
Sentence 6: Il burro si scioglie lentamente. (The butter melts slowly.) This sentence describes the process of the butter melting, indicating the action happening over time.
Sentence 7: Il burro era duro. (The butter was hard.) This sentence represents the initial state of the butter, which was hard.
Consider the answer choices:
Let's analyze each answer choice in relation to the context:
Answer A: Il burro e' sciolto. (The butter is melted.) This answer suggests that the current state of the butter is melted. However, it doesn't directly relate to the given context. We don't have any information about the current state of the butter.
Answer B: Il ragazzo e' aperto. (The boy is open.) This answer doesn't make logical sense as it equates the boy with being open, which is unrelated to the context.
Answer C: La porta era aperta. (The door was open.) This answer corresponds to the logical progression of events in the context. It follows the pattern established in sentences 2 and 3, where the door opens suddenly after being closed. This seems to be a reasonable continuation.
Answer D: Il cuoco e' sciolto. (The cook is melted.)

## B.4 Interaction 4

Here we explicitly instruct ChatGPT to use Italian. The answer is in Italian, but on the one hand it is an exact translation of the English, and secondly in one case one phrase it is not good Italian 'agenzia esterna', but a false friend translation of 'external agency'.

I am going to ask you to solve a grammatical problem called BLM-CausI. I will give you a list of Italian sentences (numbered from 1 to 7) , that I will call the context, and I will also give you a set of Italian sentences (identified by capital letters) that I will call the answers. I will ask you to choose among the answer sentences the one sentence that could be the next sentence in the context list of sentences. The sentences have been generated in a rule-based manner and follow both internal rules and a global patterns that runs across all sentences, hence the notion of sequence.

I am asking you to give me the correct answer, but also to describe step by step in Italian the hypotheses you are entertaining while you are solving the problem. You are not allowed to translate the sentences into English to solve the problem. (...)

Per risolvere il problema BLM-CausI con il contesto e le scelte di risposta fornite in italiano, analizziamo le informazioni passo dopo passo:

Identificazione degli elementi rilevanti: Gli elementi rilevanti in questo contesto sono "il ragazzo", "la porta", "il cuoco" e "il burro".

Analisi del contesto:

Frase 1: Il ragazzo apre la porta volontariamente. Questa frase stabilisce che il ragazzo apre volontariamente la porta.

Frase 2: La porta si apre di colpo. Questa frase descrive l'azione della porta di aprirsi improvvisamente, senza un'agenzia esterna.

Frase 3: La porta era chiusa. Questa frase fornisce lo stato iniziale della porta, che era chiusa.

Frase 4: La porta è aperta. Questa frase indica lo stato attuale della porta, che è aperta.

Frase 5: Il cuoco scioglie il burro volontariamente. Questa frase afferma che il cuoco fa sciogliere il burro attraverso un'azione volontaria.

Frase 6: Il burro si scioglie lentamente. Questa frase descrive il processo di scioglimento del burro, indicando che l'azione avviene nel tempo.

Frase 7: Il burro era duro. Questa frase rappresenta lo stato iniziale del burro, che era duro.

Considerazione delle opzioni di risposta:

Analizziamo ogni opzione di risposta in relazione al contesto:

Risposta A: Il burro è sciolto. Questa risposta suggerisce che lo stato attuale del burro è sciolto. Tuttavia, non è direttamente correlato al contesto fornito. Non abbiamo informazioni sullo stato attuale del burro.

Risposta B: Il ragazzo è aperto. Questa risposta non ha senso logico in quanto equipara il ragazzo a essere aperto, cosa non correlata al contesto.

Risposta C: La porta era aperta. Questa risposta corrisponde alla progressione logica degli eventi nel contesto. Segue il pattern stabilito nelle frasi 2 e 3, dove la porta si apre improvvisamente dopo essere stata chiusa. Questa sembra essere una continuazione ragionevole.

Risposta D: Il cuoco è sciolto. Questa risposta equipara il cuoco a essere sciolto, cosa che non ha senso nel contesto.

Risposta E: Il burro era sciolto. Questa risposta corrisponde alla progressione degli eventi nel contesto. Segue il pattern stabilito nelle frasi 5 e 6, dove il cuoco fa sciogliere il burro lentamente. Anche questa sembra una continuazione plausibile.

Decisione finale:

Confrontando le risposte C ed E, entrambe potrebbero essere continuazioni ragionevoli basate sul contesto fornito. Tuttavia, considerando il pattern globale e la nozione di sequenza menzionata nella descrizione del problema, sembra più probabile che la frase successiva continui a parlare dello stato della porta. Pertanto, la risposta corretta è C: La porta era aperta.

In sintesi, seguendo l'analisi passo dopo passo, abbiamo determinato che la risposta corretta è C: La porta era aperta, come continuazione più appropriata del contesto fornito.

## B.5  Interaction 5

Here we explicitly instruct ChatGPT in Italian (with a translation of the previous one).

Ti chiederò di risolvere un quesito grammaticale chiamato BLM-CausI. Ti fornirò una lista di frasi in italiano (numerata da 1 a 7), che chiameremo contesto, e ti darò anche un insieme di frasi in italiano (identificate da lettere maiuscole), che chiameremo risposte. Ti chiederò di scegliere tra le frasi di risposta quella che potrebbe essere la prossima frase nella lista del contesto. Le frasi sono state generate in base a regole e seguono sia regole interne che modelli globali che si applicano a tutte le frasi, da qui la nozione di sequenza.
Ti chiedo di darmi la risposta corretta, ma anche di descrivere passo dopo passo, in italiano, le ipotesi che stai considerando durante la risoluzione del problema. Non ti è consentito tradurre le frasi in inglese per risolvere il problema.
(...)

Per risolvere il problema BLM-CausI, analizziamo passo dopo passo le informazioni fornite:
Identificazione degli elementi rilevanti: Gli elementi rilevanti in questo contesto sono "il ragazzo", "la porta", "il cuoco" e "il burro".
Analisi del contesto:
Frase 1: Il ragazzo apre la porta volontariamente. Questa frase ci indica che il ragazzo apre la porta in modo volontario.
Frase 2: La porta si apre di colpo. Questa frase descrive l'azione della porta che si apre improvvisamente, senza un'azione esterna.
Frase 3: La porta era chiusa. Questa frase ci informa dello stato iniziale della porta, che era chiusa.
Frase 4: La porta è aperta. Questa frase indica lo stato attuale della porta, che è aperta.
Frase 5: Il cuoco scioglie il burro volontariamente. Questa frase afferma che il cuoco fa sciogliere il burro attraverso un'azione volontaria.
Frase 6: Il burro si scioglie lentamente. Questa frase descrive il processo di scioglimento del burro, indicando che avviene gradualmente nel tempo.
Frase 7: Il burro era duro. Questa frase rappresenta lo stato iniziale del burro, che era duro.
Considerazione delle opzioni di risposta:
Analizziamo ogni opzione di risposta in relazione al contesto:
Risposta A: Il burro è sciolto. Questa risposta suggerisce che il burro è attualmente sciolto, ma non abbiamo informazioni specifiche sullo stato attuale del burro nel contesto.
Risposta B: Il ragazzo è aperto. Questa risposta non ha senso logico, in quanto "aperto" si riferisce a uno stato fisico che non può essere applicato a una persona come il ragazzo.
Risposta C: La porta era aperta. Questa risposta segue la logica del contesto, in cui la porta si apre di colpo dopo essere stata chiusa. Quindi, è una continuazione plausibile.
Risposta D: Il cuoco è sciolto. Questa risposta non ha senso logico, in quanto "sciolto" si riferisce a uno stato fisico che non può essere applicato al cuoco.
Risposta E: Il burro era sciolto. Questa risposta corrisponde alla progressione logica degli eventi nel contesto. Dopo che il cuoco ha sciolto il burro volontariamente, il burro si scioglie lentamente. Quindi, questa è una continuazione coerente.
Decisione finale:
Confrontando le risposte C ed E, entrambe sembrano essere continuazioni plausibili basate sul contesto fornito. Tuttavia, considerando il pattern globale e la nozione di sequenza menzionati nella descrizione del problema, sembra più probabile che la frase successiva si riferisca allo stato della porta. Pertanto, la risposta corretta è C: La porta era aperta.
In sintesi, seguendo l'analisi passo dopo passo, abbiamo determinato che la risposta corretta è C: La porta era aperta, come continuazione più appropriata del contesto fornito.

# C  Sequence of Tenses

N.B. The interactions are formatted for readability. In the prompt, we provide the context and answer examples only for those interactions where they change with respect to the previous interaction. ChatGPT repeats the context and answer sentences literally at every interaction. For readability we have introduced ellipses, indicated with (...).

## C.1  Interaction 1

I am going to ask you to solve a grammatical problem called BLM-ConsecutioI. I will give you a list of Italian sentences (numbered from 1 to 7), that I will call the context, and I will also give you a set of Italian sentences (identified by capital letters) that I will call the answers. I will ask you to choose among the answer sentences the one sentence that could be the next sentence in the context list of sentences. The sentences have been generated in a rule-based manner.

I am asking you to give me the correct answer, but also to describe step by step the hypotheses you are entertaining while you are solving the problem.

Context
1 Se fossi ricco, allora sarei felice.
2 Se fossero buoni, allora non sarebbero ricchi.
3 Se fosti stata ricca, allora non saresti stata felice.
4 Se foste state buone, allora sareste state ricche.
5 Se sarai bravo, allora avrai un gelato.
6 Se sarete state brave, allora avrete un gelato.
7 Se sarai stato buono, allora avrai un gelato.

Answers
A Se fosti stata buona, allora avresti avuto un gelato.
B Se fosti stata buona, allora avrai un gelato.
C Se fosti stata buona, allora avranno avuto un gelato.
D Se fosti stata buona, allora sarebbero stati ricchi.
E Se fosti stata buona, allora fossero stati felici.

---

ChatGPT

To solve the grammatical problem, BLM-ConsecutioI, we need to find the sentence from the given answers that could logically follow the context sentences. Let's analyze the context and go through the hypotheses step by step.

Context: (...)

Let's analyze each answer sentence and see if it fits the context:

A. (..) This sentence is a grammatically correct conditional statement, but it doesn't fit the context. The previous sentences discuss the condition of being rich, and this answer sentence introduces a different topic about having an ice cream.

B. (..) This sentence is a grammatically correct conditional statement and it aligns with the previous context sentences. It continues the pattern of discussing the condition of being good and the consequence of having an ice cream.

C. (..) This sentence doesn't fit the context. It introduces a different subject "they" and implies that someone other than the person being addressed would have had an ice cream.

D. (..) This sentence doesn't fit the context. It introduces a different topic of being rich, which is not relevant to the previous context sentences.

E. (..) This sentence doesn't fit the context. It introduces a different topic of being happy, which is not relevant to the previous context sentences.

Based on the analysis, the correct answer is B. Se fosti stata buona, allora avrai un gelato. (If you had been good, then you will have an ice cream.) This sentence continues the pattern of discussing the condition of being good and the consequence of having an ice cream, which is consistent with the context sentences.

## C.2 Interaction 2

I am going to ask you to solve a grammatical problem called BLM-ConsecutioI. I will give you a list of Italian sentences (numbered from 1 to 7), that I will call the context, and I will also give you a set of Italian sentences (identified by capital letters) that I will call the answers. I will ask you to choose among the answer sentences the one sentence that could be the next sentence in the context list of sentences. The sentences have been generated in a rule-based manner.
I am asking you to give me the correct answer, but also to describe step by step the hypotheses you are entertaining while you are solving the problem. I am also giving you a hint: This is a grammatical problem, not a logical problem. The solution does not lie in finding a logical continuation, but a continuation that follows the underlying grammatical pattern.

ChatGPT

To solve the grammatical problem, BLM-ConsecutioI, we need to find the sentence from the given answers that could grammatically follow the context sentences. Let's analyze the context and go through the hypotheses step by step, considering the grammatical pattern.

Context: (...)

Now let's analyze each answer sentence and see if it fits the grammatical pattern:

A. (...) This sentence introduces a correct grammatical pattern and aligns with the previous context sentences. It maintains the conditional structure, discussing the consequence of having had an ice cream based on being good.

B. (...) This sentence also introduces a correct grammatical pattern and aligns with the previous context sentences. It maintains the conditional structure, discussing the consequence of having an ice cream based on being good.

C. (...) This sentence introduces a different subject "they" and implies that someone other than the person being addressed would have had an ice cream. It doesn't fit the grammatical pattern established in the context.

D. (...) This sentence introduces a different topic of being rich, which is not relevant to the previous context sentences. It doesn't fit the grammatical pattern established in the context.

E. (...) This sentence introduces a different topic of being happy, which is not relevant to the previous context sentences. It doesn't fit the grammatical pattern established in the context.

Based on the analysis, the correct answer is A. Se fosti stata buona, allora avresti avuto un gelato. (If you (feminine singular) had been good, then you (feminine singular) would have had an ice cream.) This sentence maintains the grammatical pattern, discussing the consequence of having had an ice cream based on being good, which is consistent with the context sentences.

## C.3 Interaction 3

If you had to describe in your own words the grammatical rule you have applied to find the solution what would you say?

ChatGPT

The grammatical rule applied to find the solution is the use of conditional sentences in Italian, specifically the "consecutio temporum" or sequence of tenses. In this context, the conditional sentences follow a consistent pattern:

The condition is expressed using the imperfect subjunctive tense (e.g., "se fossi," "se fossero").

The consequence is expressed using the conditional tense (e.g., "allora sarei," "allora avrai").

By analyzing the context sentences and the given answers, we identified that the correct answer should maintain this pattern. We looked for a sentence that continued the conditional structure, ensuring that the verb tenses matched the subject and conveyed the intended meaning. This allowed us to select the answer that adhered to the established grammatical rule and followed the sequence of tenses correctly.

# D   Mixture Agr and Cons

N.B. The interactions are formatted for readability. In the prompt, we provide the context and answer examples only for those interactions where they change with respect to the previous interaction. ChatGPT repeats the context and answer sentences literally at every interaction. For readability we have introduced ellipses, indicated with (...).

## D.1   Interaction 1

I am going to ask you to solve a grammatical problem called BLM-MixI. I will give you a list of Italian sentences (numbered from 1 to 7), that I will call the context, and I will also give you a set of Italian sentences (identified by capital letters) that I will call the answers. I will ask you to choose among the answer sentences the one sentence that could be the next sentence in the context list of sentences. The sentences have been generated in a rule-based manner.
I am asking you to give me the correct answer, but also to describe step by step the hypotheses you are entertaining while you are solving the problem. I am also giving you a hint: This is a grammatical problem, not a logical problem. The solution does not lie in finding a logical continuation, but a continuation that follows the underlying grammatical patterns.

Context
1 Se fossi ricco, allora sarei felice.
2 Se fossero buoni, allora non sarebbero ricchi.
3 Se fosti stata ricca, allora non saresti stata felice.
4 Se foste state buone, allora sareste state ricche.
5 Se sarai bravo, indipendentemente dalle circostanze, allora sarai ricco.
6 Se sarete stati bravi, nonostante tutto, allora sarete ricchi.
7 Se sarai buona, tenendo conto di tanti altri fattori, allora sarai ricca.

Answers
A Se sarete state buone, e con un po' di fortuna, allora sarete ricche.
B Se sarai buona, allora sarai felice.
C Se fosti stato buono, allora fossi felice.
D Se sarete stati buoni, allora sarete ricchi.
E Se fosti stata buona, tenendo conto di tanti altri fattori, allora fossero stati felici.
F Se sarete state buone, allora sarete ricche.

---

ChatGPT

To solve the grammatical problem and determine the next sentence in the given context, let's analyze the context and consider the grammatical pattern.

Context: (...)

Let's analyze each answer sentence and determine the one that follows the grammatical pattern:

A. (...) This sentence maintains the conditional structure and aligns with the previous context sentences, introducing the idea of luck as a factor. It fits the grammatical pattern and could be a valid continuation.

B. (...) This sentence introduces a different consequence of being good (happiness) instead of being rich, which deviates from the established grammatical pattern. It doesn't fit the context.

C. (...)  This sentence doesn't fit the grammatical pattern because it combines different verb tenses (subjunctive and indicative) that are inconsistent with the established pattern. It doesn't fit the context.

D. (...) This sentence maintains the conditional structure and aligns with the previous context sentences. It continues the pattern of discussing the consequence of being good and being rich. It fits the grammatical pattern and could be a valid continuation.

E. (...)  This sentence introduces a different subject "they" and doesn't follow the established grammatical pattern. It doesn't fit the context.

F. (...) This sentence maintains the conditional structure and aligns with the previous context sentences. It continues the pattern of discussing the consequence of being good and being rich. It fits the grammatical pattern and could be a valid continuation.

Based on the analysis, the correct answers could be A. Se sarete state buone, e con un po' di fortuna, allora sarete ricche. or F. Se sarete state buone, allora sarete ricche. These sentences maintain the grammatical pattern and align with the context by discussing the consequence of being good and being rich.

## D.2 Interaction 2

why not D?

ChatGPT

To solve the grammatical problem and determine the next sentence in the given context, let's analyze the context and consider the grammatical pattern.

Context: (...)

Apologies for the oversight. You're right, answer D. Se sarete stati buoni, allora sarete ricchi. is indeed a valid option based on the grammatical pattern and context.

The sentence D maintains the conditional structure and follows the established pattern of discussing the consequence of being good and being rich, which is consistent with the context sentences. Therefore, D. Se sarete stati buoni, allora sarete ricchi can also be considered as a correct continuation of the context.

## D.3 Interaction 3

There is only one correct answer. So between A and D which one is the correct one?

ChatGPT

To solve the grammatical problem and determine the next sentence in the given context, let's analyze the context and consider the grammatical pattern.

Context: (...)

Apologies for the confusion. Based on the given context and the grammatical pattern, the correct answer is D. Se sarete stati buoni, allora sarete ricchi. (If you (masculine plural) will have been good, then you (masculine plural) will be rich.)

This sentence aligns with the established pattern of conditional sentences and maintains the grammatical structure used in the context. It correctly conveys the consequence of being good, which is being rich. Therefore, D. Se sarete stati buoni, allora sarete ricchi is the correct answer.

# E    Quantitative Analysis

## E.1    Prompts for data creation

**AgrI**    Create one version of these two sets of sentences, with exactly the same syntactic structure for all of them.

Context
La discussione sulla scienza è iniziata più tardi del previsto.
Le discussioni sulla scienza sono iniziate più tardi del previsto.
La discussione sulle scienze è iniziata più tardi del previsto.
Le discussioni sulle scienze sono iniziate più tardi del previsto.
La discussione sulla scienza della vita è iniziata più tardi del previsto.
Le discussioni sulla scienza della vita sono iniziate più tardi del previsto.
La discussione sulle scienze della vita è iniziata più tardi del previsto.

Answers
A: Le discussioni sulla scienza e la vita sono iniziate più tardi del previsto.
B: Le discussioni sulle scienze della vita sono iniziate più tardi del previsto.
C: Le discussioni sulle scienze sono iniziate più tardi del previsto.
D: Le discussioni sulle scienze della vita è iniziata più tardi del previsto.
E: La discussione sulla scienza della vite è iniziata più tardi del previsto.
F: Le discussioni sulla scienza della vita sono iniziate più tardi del previsto.

**CausE**    Create one version of these two sets of sentences, with exactly the same syntactic structure for all of them. The verbs you use must belong to the following list: abate, advance, age, air, alter, atrophy, awake, balance, blast, blur, bum, burst, capsize, change, char, chill, clog, close, collapse, collect, compress, condense, contract, corrode, crumble, decompose, decrease, deflate, defrost, degrade, diminish, dissolve, distend, divide, double, drain, ease, enlarge, expand, . explode, fade, fill, flood, fray, freeze, frost, fuse, grow, halt, heal, heat, hush, ignite, improve, increase, inflate, kindle, light, loop, mature, melt, multiply, overturn, pop, quadruple, rekindle, reopen, reproduce, rupture, scorch, sear, short, short-circuit, shrink, shrivel, singe, sink, soak, splay, sprout, steep, stretch, submerge, subside, taper, thaw, tilt, tire, topple, triple, unfold, vary, warp.

Context
1 The boy opened the door voluntarily.
2 The door opened suddenly.
3 The door was closed.
4 The door is open.
5 The cook melted the butter voluntarily.
6 The butter melted slowly.
7 The butter was hard.

Answers
A The butter is melted.
B The boy is open.
C The door was open.
D The cook is melted.
E The butter was melted.

**MixI**    Create a version of this set of sentences, with exactly the same syntactic structure for all of them, but where the two adjectives before and after the comma, are always the same, for each version.

Context
1 Se fossi ricco, allora sarei felice.
2 Se fossero buoni, allora non sarebbero ricchi.
3 Se fossi stata ricca, allora non saresti stata felice.
4 Se foste state buone, allora sareste state ricche.
5 Se sarai bravo, indipendentemente dalle circostanze, allora sarai ricco.
6 Se sarete stati bravi, nonostante tutto, allora sarete ricchi.
7 Se sarai buona, tenendo conto di tanti altri fattori, allora sarai ricca.

Answers
A Se sarete state buone, e con un po' di fortuna, allora sarete ricche.
B Se sarai buona, allora sarai felice.
C Se fossi stato buono, allora fossi felice.
D Se sarete stati buoni, allora sarete ricchi.
E Se fossi stata buona, tenendo conto di tanti altri fattori, allora fossero stati felici.
F Se sarete state buone, allora sarete ricche.

## E.2    Prompts requesting solution of task

**AgrI**    I am going to ask you to solve a grammatical problem called BLM-AgrI. I will give you a list of sentences (numbered from 1 to 7), that I will call the context, and I will also give you a set of sentences (identified by capital letters) that I will call the answers. I will ask you to choose among the answer sentences the sentence that could be the next sentence in the context list of sentences.

I am asking you to give me the correct answer, but also to describe step by step your reasoning and the hypotheses you are entertaining while you are solving the problem. Please notice this is a grammatical problem. Please provide your answer in Italian.

**CausE**   I am going to ask you to solve a grammatical problem called BLM-CsE. I will give you a list of English sentences (numbered from 1 to 7), that I will call the context, and I will also give you a set of English sentences (identified by capital letters) that I will call the answers. I will ask you to choose among the answer sentences the one sentence that could be the next sentence in the context list of sentences. The sentences have been generated in a rule-based manner.

I am asking you to give me the correct answer, but also to describe step by step the hypotheses you are entertaining while you are solving the problem. Please notice this is a grammatical problem.

**MixI**   I am going to ask you to solve a grammatical problem called BLM-MixI. I will give you a list of Italian sentences (numbered from 1 to 7), that I will call the context, and I will also give you a set of Italian sentences (identified by capital letters) that I will call the answers. I will ask you to choose among the answer sentences the one sentence that could be the next sentence in the context list of sentences. The sentences have been generated in a rule-based manner.

I am asking you to give me the correct answer, but also to describe step by step the hypotheses you are entertaining while you are solving the problem. I am also giving you a hint: This is a grammatical problem, not a logical problem. The solution does not lie in finding a logical continuation, but a continuation that follows the underlying grammatical patterns.

### E.3   Quantitative results

Table 2: **Quantitative results AgrI**
Coding of Italian Agreement problem.
Legend of coding:
ItemId:a;
Properties of dataset: b:Lexical Type c:Simple/Mixed d:Repeat/Not R e:Language;
Solutions: f:Correct Answer g:Id'd Pattern;
Correctly identified elements: h:NP1 i:NP2 j:NP3 k:V;
Correctly identified attributes: l:Singular m:Plural n:Match o:Mismatch;
Organisation of Mentions: p:One-at-a-Time q:Pairwise/Grouped r:Row/Col-wise s:Unordered;
Finding Answers: t: Rule-b generation u: Answer elimination;
Comments.

| a | b | c | d | e | f | g | h | i | j | k | l | m | n | o | p | q | r | s | t | u | |
|---|---|---|---|---|---|---|---|---|---|---|---|---|---|---|---|---|---|---|---|---|---|
| e1 | I | S | NR | It | 1 | 1 | 1 | 1 | 0 | 1 | 1 | 1 | 0 | 0 | 1 | 0 | 0 | 0 | 1 | 1 | |
| e2 | I | S | NR | It | 1 | 1 | 1 | 1 | 1 | 1 | 1 | 1 | 1 | 1 | 1 | 0 | 1 | 0 | 1 | 1 | Amazingly good reasoning. |
| e3 | I | S | NR | It | 1 | 1 | 1 | 1 | 1 | 1 | 1 | 1 | 1 | 1 | 1 | 0 | 0 | 0 | 1 | 1 | |
| e4 | I | S | NR | It | 1 | 1 | 1 | 1 | 1 | 1 | 1 | 1 | 1 | 1 | 1 | 0 | 0 | 0 | 1 | 1 | |
| e5 | I | S | NR | It | 1 | 1 | 1 | 1 | 1 | 1 | 1 | 1 | 1 | 1 | 1 | 0 | 0 | 0 | 1 | 1 | Last three solutions are identical. |
| e5' | I | S | NR | It | 1 | 1 | 1 | 1 | 1 | 1 | 1 | 1 | 1 | 1 | 1 | 0 | 0 | 0 | 1 | 1 | Recognises changed pattern pl-sg-pl. |
| e1 | III | S | NR | It | 0 | 0 | 1 | 1 | 1 | 1 | 1 | 1 | 0 | 0 | 1 | 0 | 1 | 0 | 1 | 1 | |
| e2 | III | S | NR | It | 0 | 0 | 1 | 1 | 1 | 1 | 1 | 1 | 1 | 1 | 0 | 1 | 1 | 0 | 1 | 1 | if NP3 not identified, majority answer |
| e3 | III | S | NR | It | 0 | 0 | 1 | 1 | 0 | 1 | 1 | 1 | 1 | 1 | 0 | 1 | 0 | 1 | 1 | 1 | |
| e3' | III | S | NR | It | 1 | 1 | 1 | 1 | 0 | 1 | 1 | 1 | 0 | 0 | 1 | 0 | 1 | 0 | 0 | 1 | Changed prompt a bit. |
| e4 | III | S | NR | It | 1 | 1 | 1 | 1 | 1 | 0 | 1 | 1 | 1 | 1 | 0 | 1 | 0 | 1 | 1 | 1 | |
| e5 | III | S | NR | It | 0 | 1 | 1 | 1 | 0 | 1 | 1 | 1 | 1 | 1 | 1 | 0 | 0 | 0 | 1 | 1 | |

Table 3: **Quantitative results CausE**
Coding of English causative alternation problem.
Legend of coding:
a:ItemId;
Properties of dataset: b:Lexical Type c:Simple/Mixed d:Repeat/Not R e:Language;
Solutions: f: Correct Answer g: Correct Pattern h:Causative Pattern;
Correctly identified elements: i: Subject Trans1 j: Object Trans1 k: Subj Intr 1 l: Subject Trans2 m: Object Trans2 ;
Correctly identified attributes: n: Subj Intr2 o: Volition p: External Agent q: CoS: Initial r: CoS: Current s: Agent t: Patient;
Organisation of Mentions: u:One-at-a-Time v:Pairwise/Grouped w:Row/Col-wise x:Unordered;
Finding Answers: y: Rule-b generation z: Answer elimination;
Comments.

| a | b | c | d | e | f | g | h | i | j | k | l | m | n | o | p | q | r | s | t | u | v | w | x | y | z | |
|---|---|---|---|---|---|---|---|---|---|---|---|---|---|---|---|---|---|---|---|---|---|---|---|---|---|---|
| e1 | I | S | R | E | 1 | 0 | 0 | 1 | 1 | 1 | 1 | 1 | 1 | 1 | 0 | 1 | 1 | 0 | 0 | 0 | 1 | 1 | 0 | 1 | 1 | This is a grammatical problem at end of instructions, very 'grammatical' solution. |
| e1' | I | S | R | E | 1 | 0 | 0 | 1 | 1 | 1 | 1 | 1 | 1 | 1 | 0 | 1 | 1 | 0 | 0 | 1 | 0 | 1 | 0 | 1 | 1 | No 'this is a grammatical problem' at the end of instructions more semantic solution. |
| e2 | I | S | R | E | 1 | 0 | 0 | 1 | 1 | 1 | 1 | 1 | 1 | 0 | 0 | 1 | 1 | 0 | 0 | 0 | 1 | 0 | 0 | 1 | 1 | |
| e3 | I | S | R | E | 1 | 0 | 0 | 1 | 1 | 1 | 1 | 1 | 1 | 0 | 0 | 0 | 0 | 0 | 0 | 0 | 1 | 1 | 0 | 1 | 1 | |
| e4 | I | S | R | E | 1 | 0 | 0 | 1 | 1 | 1 | 1 | 1 | 1 | 0 | 0 | 0 | 0 | 0 | 0 | 0 | 1 | 1 | 0 | 1 | 1 | |
| e5 | I | S | R | E | 1 | 0 | 0 | 1 | 1 | 1 | 1 | 1 | 1 | 0 | 0 | 0 | 0 | 0 | 0 | 0 | 1 | 1 | 0 | 1 | 1 | |
| e1 | III | S | R | E | 1 | 0 | 0 | 0 | 1 | 1 | 0 | 1 | 1 | 1 | 0 | 1 | 1 | 0 | 0 | 0 | 1 | 1 | 0 | 0 | 1 | Makes a mistake in the analysis, thinks it's a passive active alternation |
| e2 | III | S | R | E | 1 | 1 | 0 | 0 | 1 | 1 | 0 | 1 | 1 | 0 | 0 | 1 | 1 | 0 | 0 | 0 | 1 | 1 | 0 | 1 | 1 | Again some grammatical analysis mistakes but pretty cool analysis |
| e3 | III | S | R | E | 1 | 0 | 0 | 0 | 1 | 1 | 0 | 1 | 1 | 0 | 0 | 1 | 1 | 0 | 0 | 0 | 1 | 1 | 0 | 1 | 1 | Again rules are not correct |
| e4 | III | S | R | E | 1 | 0 | 0 | 0 | 1 | 1 | 0 | 1 | 1 | 1 | 0 | 1 | 1 | 0 | 0 | 0 | 1 | 1 | 0 | 1 | 1 | Same as above, exactly same reasoning. |
| e5 | III | S | R | E | 1 | 0 | 0 | 0 | 1 | 1 | 0 | 1 | 1 | 0 | 0 | 1 | 1 | 0 | 0 | 0 | 1 | 1 | 0 | 1 | 1 | Same, attempt rules but not correct, very superficial. |

Table 4: **Quantitative results MixI**
Coding of Italian Mixed Agreement and Sequence of Tense problem.
Legend of coding:
a:ItemId;
Properties of dataset: b:Lexical Type c:Simple/Mixed d:Repeat/Not R e:Language;
Solutions: f: Correct Answer g: Id'd Pattern;
Correctly identified elements: h: NP1 i: NP2 j: Incidental k: V l: premisse m: consequent;
Correctly identified attributes: n:Singular o:Plural p:Match q:Mismatch r:Tense s:Person t:Gender u:Match v:Mismatch;
Organisation of Mentions: w:One-at-a-Time x:Pairwise/Grouped y:Row/Col-wise z:Unordered;
Finding Answers: aa: Rule-b generation ab: Answer elimination;
Comments.

| a | b | c | d | e | f | g | h | i | j | k | l | m | n | o | p | q | r | s | t | u | v | w | x | y | z | aa | ab | |
|---|---|---|---|---|---|---|---|---|---|---|---|---|---|---|---|---|---|---|---|---|---|---|---|---|---|---|---|---|
| e1 | I | M | NR | It | 1 | 1 | 1 | 1 | 0 | 1 | 1 | 1 | 1 | 1 | 1 | 1 | 1 | 0 | 1 | 1 | 1 | 0 | 1 | 1 | 0 | 1 | 1 | |
| e2 | I | M | NR | It | 1 | 0 | 1 | 1 | 0 | 1 | 1 | 1 | 1 | 1 | 1 | 1 | 1 | 1 | 1 | 1 | 1 | 1 | 0 | 1 | 0 | 1 | 1 | Gets good answer by a strangely incorrect reasoning. |
| e3 | I | M | NR | It | 1 | 1 | 1 | 1 | 1 | 1 | 1 | 1 | 1 | 1 | 1 | 1 | 1 | 1 | 1 | 1 | 1 | 0 | 1 | 1 | 0 | 1 | 1 | |
| e4 | I | M | NR | It | 1 | 0 | 1 | 1 | 1 | 1 | 1 | 1 | 1 | 1 | 1 | 1 | 1 | 1 | 1 | 1 | 1 | 1 | 1 | 1 | 0 | 0 | 1 | |
| e5 | I | M | NR | It | 1 | 1 | 1 | 1 | 1 | 1 | 1 | 1 | 1 | 1 | 1 | 1 | 1 | 1 | 1 | 1 | 1 | 0 | 1 | 1 | 0 | | | Despite making mistakes in grammatical analysis of sing, plural, it sees the pattern. |
| e1 | III | M | NR | It | 0 | 0 | 0 | 0 | 0 | 0 | 1 | 1 | 1 | 1 | 0 | 0 | 0 | 1 | 0 | 0 | 0 | 1 | 0 | 1 | 0 | 0 | 1 | |
| e2 | III | M | NR | It | 1 | 1 | 0 | 0 | 0 | 0 | 1 | 1 | 1 | 1 | 0 | 0 | 0 | 1 | 1 | 1 | 1 | 1 | 0 | 1 | 0 | 1 | 1 | |
| e3 | III | M | NR | It | 1 | 1 | 1 | 1 | 1 | 1 | 1 | 1 | 1 | 1 | 1 | 1 | 1 | 1 | 1 | 1 | 1 | 0 | 0 | 1 | 0 | 1 | 1 | Marvellous, understood all underlying elements. |
| e4 | III | M | NR | It | 1 | 0 | 0 | 0 | 0 | 0 | 0 | 0 | 1 | 1 | 0 | 0 | 1 | 1 | 0 | 0 | 0 | 1 | 0 | 1 | 0 | 0 | 1 | |
| e5 | III | M | NR | It | 1 | 1 | 1 | 1 | 1 | 1 | 1 | 1 | 1 | 1 | 1 | 1 | 1 | 1 | 0 | 0 | 0 | 0 | 1 | 1 | 0 | 1 | 1 | Very good, like 3. |