# OpenReview forum: "Blackbird language matrices (BLM), a new task for rule-like generalization  in neural networks: Can Large Language Models pass the test?"
_EMNLP/2023/Conference — EMNLP 2023 Findings_

### Official Review · Reviewer_djfD · 2023-07-24

**Soundness:** 2

**Excitement:**

3: Ambivalent: It has merits (e.g., it reports state-of-the-art results, the idea is nice), but there are key weaknesses (e.g., it describes incremental work), and it can significantly benefit from another round of revision. However, I won't object to accepting it if my co-reviewers champion it.

**Missing References:**

In-Context Analogical Reasoning with Pre-Trained Language Models by Hu et al (ACL 2023): https://arxiv.org/abs/2305.17626


**Paper Topic And Main Contributions:**

This paper extends the existing work and qualitatively looks at ChatGPT responses to prompts that exemplify different aspects of rule-based linguistic reasoning such as correspondence finding and item novelty. While the overall idea is interesting, the paper is a collection of anecdotal evidence that lacks more robust qualitative evaluation. As well, it is unclear what human performance on this task is, and how does it compare to that of ChatGPT or other LLMs. In this form, given the level of technical detail needed to understand the manuscript, it might be of limited interest to the NLP audience without more robust evaluation.

**Questions For The Authors:**

1. How do humans perform on tasks you study?

2. Surely you would not conclude that a person can speak another language if they correctly translate one sentence into that language from their native tongue, so, by the same analogy -- how can we conclude that "ChatGPT is good at correspondence finding" based on one interaction?

3. In your ethics statement the authors state no concerns, however, one could argue that one pressing question is that of anthropomorphization of neural networks, especially in the eyes of non-technical audience. For example, they mention "evaluation of LLMs intelligent behaviour" and "[LLMs] an easily understand the instructions and provde step-by-step reasoning". Can you comment on potential issues with the choice of such expressions?


**Reasons To Accept:**

- Raven's progressive matrices are well established in decision making literature and thinking about ways to transfer this to the linguistic domain is an interesting and well motivated idea that would be of interest to wider community
- The authors include prompts in English and Italian, thus testing their hypothesis more generally and not just on one language


**Reasons To Reject:**

- Hypotheis that this kind of linguistic reasoning is somehow universal is not substantiated by empirical data from humans. Some prompts (such as the one shown in Figure 5) appear quite difficult and I am not convinced that even humans would perform well on the task.

- Overreaching claims based on anecdotal data/qualitative observations: The appendix is a collection of prompts including responses from ChatGPT. The prompts they design have interesting properties, but they are not presented or described as a benchmark, and the results should be also presented in a quantitative form to have some methodological validity

- Manuscript is not self-contained, the reader needs to constantly refer to the appendix which makes it difficult to follow. As well, I found the content hard to read due to the level of technical detail related to linguistic phenomena the authors are testing and lack of examples in the main body of the text

**Reproducibility:**

4: Could mostly reproduce the results, but there may be some variation because of sample variance or minor variations in their interpretation of the protocol or method.

**Reviewer Confidence:**

4: Quite sure. I tried to check the important points carefully. It's unlikely, though conceivable, that I missed something that should affect my ratings.

---

> ### Author Rebuttal · Authors · 2023-08-28
>
> We disagree with the reviewer’s statement that the paper requires great technical knowledge. The paper contains only  linguistic or NLP notions that are to be found in an introductory undergraduate textbook on NLP (such as for example, Jurafky and Martin),  and is voluntarily kept very accessible to the whole NLP community.  It is also not true that there are no examples in the text: Figure 5, figure 6 and figure 8 are all full examples of the BLM that were tested.
>
> It is entirely possible that some humans would not perform well on this task, these are after all IQ tests, so they might be correlated to IQ. However, we think that we would all agree that humans would be able to see the pattern, if given all the help that is given in the tests illustrated in the appendix. Informal experiments by the author in several languages have shown that humans have no trouble with lexically uniform data (such as those in figure 6 and 8), but have more trouble with lexically varied data (such as those in figure 5).
>
> The paper follows rather standard qualitative analysis procedures. What the reviewer calls ‘anecdotal’, is in fact a replication of a think-aloud protocol, as in Carpenter et al 1990, a perfectly acceptable protocol, which produces data subject to qualitative analysis, a sound scientific methodology. These are not anecdotes: the interactions use data developed based on a formal framework (BLMs), they are drawn from four different types of problems, they are drawn from two languages, and provide 27 different experimental interactions, hence 27 different experimental trial. While we can certainly do a bigger study, and we will, these are convincingly solid bases. We should also add that still impressionistically we have had many more interactions, with roughly the same results.
>
> The claims are not over-reaching, instead they are suggestive, as clearly stated in the introduction this is a conceptual piece and a proof of concept.
>
> We respectfully disagree with the statement that one cannot say that a speaker knows the language from the translation of one sentence: it depends on the length, structural and semantic complexity of the sentence. In a very complex case that covers several phenomena one could reasonably conclude that at least some aspects of the language are known.
>
> Moreover, extensive studies on other aspects of ChatGPT have been performed (see Bang et al 2023). All this conceptual paper is meant to show is that these kinds of qualitative analyses provide useful ways of evaluating the systems that complement the larger evaluations, and that BLMs provide a framework to know what to ask and how to ask it.
>
> We agree that the anthropomorphisation of LLMs is an interesting and important topic. We didn't say that ChatGPT is human, we said it appears to exhibit behaviours analogous to  human behaviours, in this limited set of tests.
>
> In our opinion, the paper the reviewer mentions cannot be considered a ‘missing reference’ according to the reviewing guidelines. The paper  came out in ACL 2023 and is, as such, to be considered contemporary or even subsequent to this submission.  We are aware of this paper and will certainly add the citation, but since the work in that paper  is fundamentally on vision, it is not directly related to the problem in this paper. It is also *not* preceding the BLM work. BLM work has been around since 2022, and published in EACL 2023 (and is not cited in the ACL 2023 paper the reviewer mentions, thus showing that paper’s lack of knowledge of the NLP literature on their own topic).

---

### Official Review · Reviewer_jZnL · 2023-07-24

**Typos Grammar Style And Presentation Improvements:** Shorten the abstract and in general t…
**Soundness:** 3

**Excitement:**

3: Ambivalent: It has merits (e.g., it reports state-of-the-art results, the idea is nice), but there are key weaknesses (e.g., it describes incremental work), and it can significantly benefit from another round of revision. However, I won't object to accepting it if my co-reviewers champion it.

**Missing References:**

none

**Paper Topic And Main Contributions:**

The paper tests to what extent LLMs, chatGPT in this case, pass a certain cognitive task called the Blackbird Language Matrice test. Similarly to visual puzzles, where a pattern needs to be discerned, a linguistic analog was crafted in this paper. One pattern, for example, is that the subject and verb have the same plurality. The overall conclusion is that chatGPT does excel in the task.

**Questions For The Authors:**

see reason to reject

**Reasons To Accept:**

This paper sheds light on how chatGPT works. There is a nice qualitative analysis of the answers, and one can conclude the way chatGPT "thinks" and what are its limitations in terms of generalization to new domains of knowledge, and in general to being creative like humans.

**Reasons To Reject:**

The test set is small (a few examples), so a statistically sound conclusion may not be drawn.
Also, a comparison to human performance is lacking. So it's not clear if the poor performance of chatGPT in some tasks would replicate in humans (adults/kids/cognitively-imapred people)?


**Reproducibility:**

5: Could easily reproduce the results.

**Reviewer Confidence:**

2: Willing to defend my evaluation, but it is fairly likely that I missed some details, didn't understand some central points, or can't be sure about the novelty of the work.

---

> ### Author Rebuttal · Authors · 2023-08-28
>
> First of all, we must start by clarifying that the results of the paper are *not* what the reviewer states. We do not claim that ChatGPT excels in the  task we evaluate. On the contrary, we show, with careful qualitative analyses, that it does *not* manage to solve the overall task. But,  we manage to disaggregate the overall negative results into finer-grained conclusions, some of which are positive. This is, in our opinion, the real interest of this approach, as it allows us to investigate the ChatGPT scene in a more articulated manner.
>
> Secondly, wrt to comparison to humans: these are tests of intelligence, as such they have been largely studied it in their visual paradigm and we understand their nature. We draw analogies here to the performance of humans precisely to demonstrate the proof of concept. We are planning comparisons with humans, but such a comparison is complex as these are intelligence tests, not natural human behaviour.  This will yield a different paper.
>
> Concerning the length of the introduction: we are happy to try to make it more concise, but we don’t share the reviewer’s opinion that it is too long. As stated in the introduction, this is a conceptual  piece, so its perceived verbosity is in reality a necessity of argumentation.

---

### Official Review · Reviewer_6RnV · 2023-08-04

**Soundness:** 4

**Excitement:**

3: Ambivalent: It has merits (e.g., it reports state-of-the-art results, the idea is nice), but there are key weaknesses (e.g., it describes incremental work), and it can significantly benefit from another round of revision. However, I won't object to accepting it if my co-reviewers champion it.

**Paper Topic And Main Contributions:**

The paper aims to study the linguistic reasoning of LLMs (specifically ChatGPT) by prompting them to solve simple tasks from a language task termed Blackbird language matrices. The language task is analogous to Raven's Progressive Matrices for Vision, aiming to study correspondence, novelty, and sub-goal management. The paper argues that LLMs (specifically ChatGPT) display linguistic reasoning to a certain extent and are successful with correspondence and novelty-based tasks, however, they tend to fail at sub-goal management. The paper aims at establishing a benchmark that could potentially pave the way for studying linguistic reasoning ability in LLMs as well as prompt the community toward developing more such analogous benchmarks.

**Questions For The Authors:**

Over time, with some efforts toward building an evaluation metric, and adding more baselines, this paper could really come to stand out. With the current state of the paper, however, this would be more suited to a workshop venue.

**Reasons To Accept:**

1. Very well-written, easy-to-follow, comprehensive set of qualitative experiments.
2. Using the BLM seems like an ideal way forward toward studying the reasoning ability of language models through something as fundamental as language. This of course also serves as a stepping stone toward building other such reasoning tasks (Raven's Progressive Matrices, Blackbird Language Matrices, and more to come).
3. The qualitative assessments seem to have done a very good job of analyzing the linguistic reasoning ability of ChatGPT with a thorough analysis of how it reasons with respect to correspondence, novelty and subgoal management.

**Reasons To Reject:**

1. Lack of Quantitative Evaluation/Metric: The paper seems to have offered a very thorough qualitative evaluation, however, going forward, a purely qualitative evaluation might limit the benchmark's ability to be tested over a breadth of models and puzzles.
2. Lack of Baselines: The paper *ONLY* uses ChatGPT as the LLM to evaluate the benchmark. One would hope to see this tested against other state-of-the-art LLMs as well.
3. Incremental Work: This work is incremental compared to other previous works like i) Blackbird's language matrices (BLMs): a new benchmark to investigate disentangled generalisation in neural networks. ii) Blackbird language matrices (BLM), a new task for rule-like generalization in neural networks: Motivations and Formal Specifications. iii) BLM-AgrF: A New French Benchmark to Investigate Generalization of Agreement in Neural Networks.

**Reproducibility:**

N/A: Doesn't apply, since the paper does not include empirical results.

**Reviewer Confidence:**

4: Quite sure. I tried to check the important points carefully. It's unlikely, though conceivable, that I missed something that should affect my ratings.

---

> ### Author Rebuttal · Authors · 2023-08-28
>
> We believe there is a certain misunderstanding on the nature and the goals of this paper. This paper was submitted to the theme track, which accepts conceptual papers. As clearly stated in the abstract and introduction, this is a ‘conceptual’ paper and it offers the proof of concept  that the method proposed in BLMs could work as an evaluation of ChatGPT. It offers a proof of concept, and as such, we  believe, convincingly shows that the method is useful to break down more global claims of reasoning and understanding into more specific abilities, and it relates them to language.
>
> We do not share the opinion that a quantitative evaluation is intrinsically better than a qualitative one. While we agree that for a different kind of paper more data could be interesting, we believe we have provided an amount and variety of data points that convincingly support the proof of concept and justify future larger evaluation campaigns using this method.
>
> Going forward, more interactions can be accumulated and analyzed, but we think that a qualitative approach is really necessary here, even if quantified on a larger sample. A quantitative approach instead would measure quantitative properties of the interactions, which we don’t think would be insightful, at the current stage of knowledge of reasoning abilities on these tests and would invariably simply repeat what has already been done (see for example the excellent Bang et al 2023).
>
>
> Again, this paper offers a proof of concept, not a scientific contribution evaluating reasoning in all language models. The paper intentionally sets forward to evaluate ChatGPT, it does not make any claims of general validity other than for  motivating the approach. We are interested in ChatGPT as the model that most people in the world have been discussing. We don’t agree that this is an unsound scientific process.
>
>
> We emphatically disagree that the work is  incremental. It uses a novel data framework and evaluates it in a completely different setting than what was done before. Nothing of what constitutes the core contribution of the work, other than the obviously necessary presentation of the framework, has been done before.

---

### Meta-Review · Area_Chair_MD1o · 2023-09-15

**Recommendation:** 3

**Metareview:**

The reviewers gave scores 4,3,2 (soundness) and 3,3,3 (excitement).

The following strengths and weaknesses were prominent:

Strengths:

- well-written (R1)
- link to well-established idea (Raven's matrices) from psychology (R3)
- comprehensive experiments (R1)
- interesting qualitative analysis (R1, R3)
- includes two languages (R3)

Weaknesses

- lack of quantitative evaluation (R1, R3) and small test set (R2)
- lack of baselines, narrowly focusing on ChatGPT (R2)
- perceived as incremental (R2)
- unclear whether this kind of reasoning is accessible to humans (R3)
- main text is not self-contained (R3)

On reconciling the disparate scores:

R3 (Reviewer djfD) gave the lowest soundness score. One of the key weakness they mentioned, such as lack of quantitative evaluation, were also noted by the other reviewers, including R1 (6RnV), who nonetheless gave a high (4) soundness score. The other key weakness mentioned by R3, and not so much by the others is doubt about whether this kind of reasoning is actually accessible to humans. On the other hand, the link to Raven's matrices, a well-established idea in psychology was mentioned positively.

Taken together, an overall lack of quantitative evaluation appears to be the main soundness weakness noted by all three reviewers, including those giving high (4), medium (3), and low (2) soundness scores.

---

### Decision · Program_Chairs · 2023-10-07

**Decision:**

Accept-Findings

**Comment:**

The reviewers gave scores 4,3,2 (soundness) and 3,3,3 (excitement).

The following strengths and weaknesses were prominent:

Strengths:

- well-written (R1)
- link to well-established idea (Raven's matrices) from psychology (R3)
- comprehensive experiments (R1)
- interesting qualitative analysis (R1, R3)
- includes two languages (R3)

Weaknesses

- lack of quantitative evaluation (R1, R3) and small test set (R2)
- lack of baselines, narrowly focusing on ChatGPT (R2)
- perceived as incremental (R2)
- unclear whether this kind of reasoning is accessible to humans (R3)
- main text is not self-contained (R3)

On reconciling the disparate scores:

R3 (Reviewer djfD) gave the lowest soundness score. One of the key weakness they mentioned, such as lack of quantitative evaluation, were also noted by the other reviewers, including R1 (6RnV), who nonetheless gave a high (4) soundness score. The other key weakness mentioned by R3, and not so much by the others is doubt about whether this kind of reasoning is actually accessible to humans. On the other hand, the link to Raven's matrices, a well-established idea in psychology was mentioned positively.

Taken together, an overall lack of quantitative evaluation appears to be the main soundness weakness noted by all three reviewers, including those giving high (4), medium (3), and low (2) soundness scores.